# TransCues: Boundary and Reflection-empowered Pyramid Vision Transformer for Semantic Transparent Object Segmentation

## Abstract

Although glass is a prevalent material in everyday life, most semantic segmentation methods struggle to distinguish it from opaque materials. We propose **TransCues**, a pyramidal transformer encoder-decoder architecture to segment transparent objects from a color image. To distinguish between glass and non-glass regions, our transformer architecture is based on two important visual cues that involve boundary and reflection feature learning, respectively. We implement this idea by introducing a Boundary Feature Enhancement (BFE) module paired with a boundary loss and a Reflection Feature Enhancement (RFE) module that decomposes reflections into foreground and background layers. We empirically show that these two modules can be used together effectively, leading to improved overall performance on various benchmark datasets. In addition to binary segmentation of glass and mirror objects, we further demonstrate that our method works well for generic semantic segmentation for both glass and non-glass labels. Our method outperforms the state-of-the-art methods by a large margin on diverse datasets, achieving **+4.2%** mIoU on Trans10K-v2, **+5.6%** mIoU on MSD, **+10.1%** mIoU on RGBD-Mirror, **+13.1%** mIoU on TROSD, and **+8.3%** mIoU on Stanford2D3D, demonstrate the effectiveness and efficiency of our method.

## 1 Introduction

Glass, present in diverse forms like windows, bottles, and walls, poses a significant challenge for image segmentation due to its appearance heavily influenced by the surrounding environment. While many robotic systems (Xie et al., 2020; 2021b; Zhang et al., 2022) often rely on sensor fusion techniques, such as sonars or lidars, these often falter when it comes to detecting transparent objects, causing scan matching problems. The unique properties of transparent objects, such as refraction and reflection, can deceive robot sensors. For collision prevention, robots in environments like workplaces, supermarkets, or hotels need to accurately recognize glass barriers. Similarly, in domestic and professional settings, robots must navigate around fragile objects like vases and glasses. Thus, a practical robust cost-effective vision-based approach for transparent object segmentation is highly desirable. Despite such, the majority of existing semantic segmentation algorithms (Xie et al., 2021a; Zheng et al., 2021; Yang et al., 2018; Chen et al., 2018; Yuan et al., 2021b) do not adequately account for transparent and reflective objects, leading to a notable decline in their performance when such objects are present in a scene. For example, when there are objects that reflect light, these algorithms might misinterpret the image of the reflective objects as real objects. As a result, this will degrade the performance of different 2D and 3D computer vision tasks, such as robot navigation, depth estimation, 3D reconstruction, etc.

To solve this problem, this paper introduces an *efficient transformer-based architecture* tailored for segmenting **transparent** and **reflective** objects along with **general** objects. Our approach harnesses the power of transformer-based encoders and decoders, allowing us to capture long-range contextual information, unlike previous methods, which relied heavily on stacked attention layers (Fu et al., 2019; Yang et al., 2021) or combined CNN backbones with transformers (Xie et al., 2021b; Zheng et al., 2021; Wang et al., 2021b). These long-range visual cues are essential to reliably identify transparent objects, especially when they lack distinctive textures or share similar content with their surroundings (Xie et al., 2021b). To build our transformer, we take inspiration from two visual cues

in human perception, which employ boundary localization for shape inference and reflections for glass surface recognition. Our method combines a **geometric cue** based on glass boundaries and an **appearance cue** based on glass reflections to enhance feature learning in our network. **First**, we introduce a Boundary Feature Enhancement (BFE) module to learn and integrate glass boundary features that can help in both localization and segmentation of the glass-like regions. We self-supervise this module by a new boundary loss that utilizes the Sobel kernel to extract boundaries based on the gradients of the predictions and ground-truth objects' masks. **Second**, we introduce a Reflection Feature Enhancement (RFE) module, which decomposes reflections into foreground and background layers, providing the network with additional features to distinguish between glass-like and non-glass areas. More importantly, we demonstrate that our method is robust to both transparent object segmentation and generic semantic segmentation tasks, with state-of-the-art performance for both scenarios across various datasets.

In summary, our contributions are as follows: **(1)** We introduce TransCues, an efficient transformer-based segmentation architecture that segments both transparent, reflective, and general objects. **(2)** We propose the Boundary Feature Enhancement (BFE) module and boundary loss, improving the accuracy of glass detection performance. We present the Reflection Feature Enhancement (RFE) module, facilitating the differentiation between glass and non-glass regions. **(3)** We demonstrate the effectiveness of both modules for generic semantic segmentation. **(4)** We conducted extensive experiments and ablation studies that demonstrate the state-of-the-art performance of our method.

## 2 RELATED WORKS

### 2.1 TRANSPARENT OBJECT AND MIRROR SEGMENTATION

**Transparent Object Segmentation.** For transparency, the intensity of both glass and background often matches, making it challenging to differentiate. Traditional visual aid systems, enhanced with ultrasonic sensors and RGB-D cameras, effectively identify transparent barriers like glass and windows (Huang et al., 2018). Following works explored the use of transmission differences (Okazawa et al., 2019), reflection cues (Lin et al., 2021), and polarization (Xiang et al., 2021; Mei et al., 2022) for detecting transparency. Moreover, methods developed for transparency segmentation (Xie et al., 2021b; He et al., 2021; Mei et al., 2023) cater to a range of objects from opaque entities like windows and doors to see-through items such as cups and eyeglasses, focusing on discerning reflections and their boundaries to accurately detect and define transparent surfaces.

The Trans10K-v2 dataset by (Xie et al., 2021b) has revealed the potential of RGB-based segmentation for transparent objects, prompting new research directions beyond conventional sensor fusion. This includes AdaptiveASPP (Cao et al., 2021) for enhanced feature extraction and EBLNet (He et al., 2021) for improved global form representation. Alongside these developments, Trans4Trans (Zhang et al., 2022) is proposed to provide a lightweight, general network for real-world applications. Our work, drawing on these innovations, aims to create an efficient, robust solution for transparent object segmentation, suitable for general semantic segmentation and practical uses like robot navigation.

**Mirror Segmentation.** In the field of mirror segmentation, recent models have introduced high-level concepts to improve detection and localization. SANet (Guan et al., 2022) utilizes the semantic relationships between mirrors and their surrounding environment for precise localization. SAT-Net (Huang et al., 2023) capitalizes on the natural symmetry between objects and their mirror reflections to accurately identify mirror locations. VCNet (Tan et al., 2023), on the other hand, explores 'visual chirality'—a unique property of mirror images—and incorporates this through a specialized transformation process for effective mirror detection. Lastly, HetNet (He et al., 2023) introduces a unique model combining multi-orientation intensity-based contrasted (MIC) modules for initial mirror localization using low-level features, and reflection semantic logical (RSL) modules for high-level semantic analysis.

### 2.2 TRANSFORMER IN SEMANTIC SEGMENTATION

Previously, the pioneer Fully Convolutional Networks (FCNs) (Long et al., 2015) could complete semantic segmentation end-to-end by treating it as a dense pixel classification task. Modern techniques build on this paradigm by enhancing FCNs with context aggregation modules. To expand receptive fields, PPM (Zhao et al., 2017) utilizes various scales of pooling operators in a pyramidal fashion,

whereas ASPP (Chen et al., 2018) uses atrous convolutions with varying dilation rates. DANet (Fu et al., 2019), OCNet (Yuan et al., 2021b), and CCNet (Huang et al., 2019) all use different kinds of non-local attention blocks to take advantage of long-range relationships between pixels. Disentangled and asymmetric variants of non-local modules (Zhu et al., 2019; Yang et al., 2021) have also been developed to minimize the computational complexity of dense pixel-pair connections.

Since its introduction in natural language processing, transformers have been adopted and further investigated for computer vision tasks. One of the pioneers is Vision Transformer (ViT) (Dosovitskiy et al., 2021), which applies transformer layers to sequences of image patches. SETR (Zheng et al., 2021) and Segmenter (Strudel et al., 2021) take inspiration from ViT and directly add upsampling and segmentation heads to learn long-range context information from the initial layer. MaX-DeepLab (Wang et al., 2021a), and MaskFormer (Cheng et al., 2021) study 2D image segmentation through the perspective of masked prediction and classification based on recent advances of object detection using transformers (Carion et al., 2020). As a result, several transformer-based methods for dense image segmentation have been developed (Liu et al., 2021b; Xie et al., 2021a). Pyramid architectures of vision transformers have been proposed by PVT (Wang et al., 2021b) and SegFormer (Xie et al., 2021a) as a method for gathering hierarchical feature representations. Both ECANet (Yang et al., 2021) and CSWin transformer (Dong et al., 2022) recommend applying a self-attention mechanism in either vertical or horizontal stripes to gain advanced simulation capacity while minimizing computing overheads. NAT (Hassani et al., 2023), on the other hand, aims to simplify the standard attention mechanism, resulting in faster processing and reduced memory requirements.

Recent methods in computer vision have been trying to match the performance of transformer models using advanced CNN architectures. MogaNet (Li et al., 2023b) introduces two feature mixers with depthwise convolutions, efficiently processing middle-order information across spatial and channel spaces. InternImage (Wang et al., 2023) utilizes deformable convolution, providing a large effective receptive field essential for tasks like detection and segmentation, and offers adaptive spatial aggregation based on input and task specifics. These approaches collectively signify a shift towards more efficient, task-tailored CNN models that strive to replicate the success of transformers in various computer vision applications.

# 3 OUR PROPOSED METHOD

## 3.1 OVERVIEW

Our network is built upon visual cues from transparent objects. We focus on two important visual cues: boundaries between glass and non-glass regions, and glass reflections. Detecting boundaries is often more straightforward than identifying reflections because reflections can be view-dependent, and so glass surfaces may not always exhibit strong reflections. Boundaries frequently indicate high-contrast edges around transparent objects, aligning with human visual perception. For this reason, we initially employed the BFE module to enhance the boundary characteristics of transparent features. Subsequently, we utilize the enhanced features as input to the RFE module. These two modules are built upon a strong feature learning backbone using a pyramid vision transformer.

As depicted in Figure 1, our network's architecture adheres to the well-established encoder-decoder structure, comprising the Feature Extraction Module (FEM), Feature Parsing Module (FPM), Boundary Feature Enhancement (BFE), and Reflection Feature Enhancement (RFE) modules. Precisely, the FEM module, based on the PVT architecture (Wang et al., 2022) in the encoder, efficiently captures multi-scale long-range dependency features from the input image. Drawing inspiration from (Zhang et al., 2022), the FPM module offers a lightweight alternative to the FEM module. To enrich the feature learning capabilities of our network, we leverage a geometric cue based on glass boundaries (BFE module) and an appearance cue based on glass reflections (RFE module). These cues enhance our network's ability to capture fine details and long-range contextual information for transparent features. Consequently, as the image data progresses through our decoder, it integrates contextual information of varying resolutions, preserving the fine-grained information of transparent features. Finally, a compact MLP layer is employed to predict semantic labels for each pixel.

Given an RGB image, represented as $\mathbf{I} \in \mathbb{R}^{H \times W \times 3}$, where $H$ and $W$ represent the image height and width, respectively, our network aims to segment this image into semantic labels for each pixel, which can be expressed as $\mathbf{F} \in \mathbb{R}^{H \times W \times n_{class}}$, where $n_{class}$ represents the number of classes. To handle

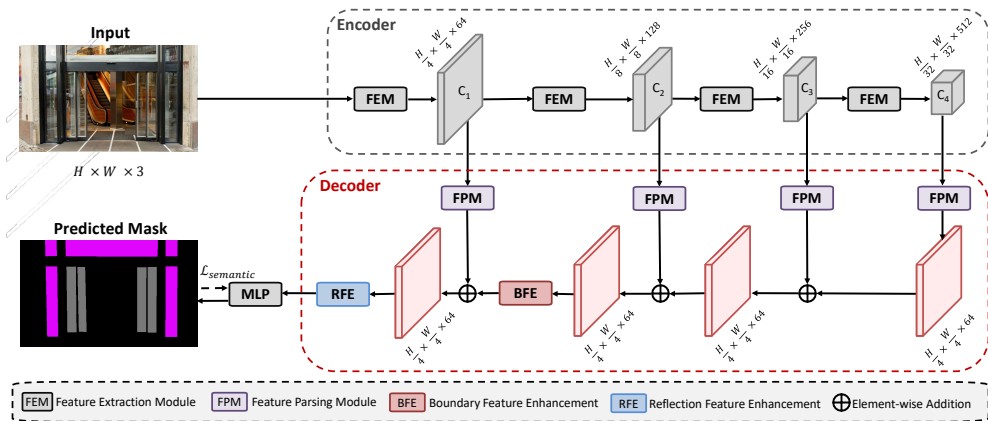

Figure 1: **Overview of our TransCues method.** An RGB image is processed by four FEM modules in the encoder for multi-scale feature extraction. These features are then refined by the decoder's FPM, BFA, and RRA modules, culminating in semantic labels via an MLP (see Figure 2). Our main contributions, BFA and RRA modules, are elaborated in Sections 3.2 and 3.3.

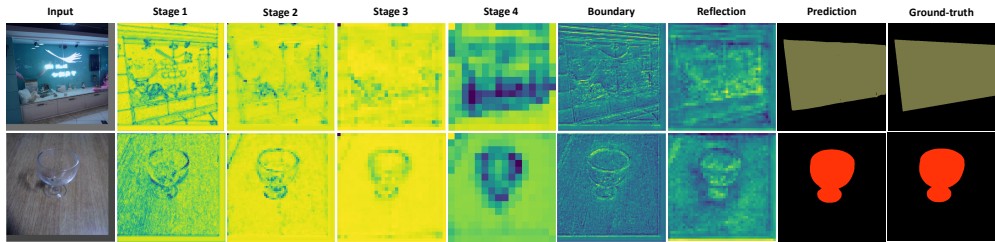

Figure 2: Visualization of feature maps of our method. Zoom in for better visualization.

variations in input image sizes across datasets, we standardize the resolution to either $512 \times 512$ or $768 \times 768$, ensuring consistent position embedding dimensions throughout both training and testing phases. The network components are detailed as follows.

**Position Embedding.** We embed position information into the PVT-inspired architecture (Wang et al., 2021b), adhering to the ViT (Dosovitskiy et al., 2021) protocol of using randomly initialized position parameters. To overcome non-transferable pre-trained position embeddings due to different image dimensions, we adapt the resolution via bilinear interpolation of ImageNet-derived embeddings.

**Encoder and Decoder.** Our encoder-decoder setup, taking cues from PVT (Wang et al., 2021b), employs a pyramidal transformer design for multi-scale feature embeddings, enhancing segmentation performance. Our adaptable framework supports various transformer backbones (see Section 4.2), including PVTv2 (Wang et al., 2022) for its efficient feature extraction and memory usage. For segmentation, we discard PVTv2's classifier and introduce an FPM module over the multi-level features from the FEM module, capturing detailed to abstract representations of transparent objects across $C_1$ to $C_4$. Please see the appendix Figure 6 for their artchitecture.

As shown in Figure 2, throughout each stage in the encoder and the boundary and reflection modules, our feature maps show that the glass region, including its defining boundary and reflection, is clearly distinguishable from non-glass surfaces.

## 3.2 BOUNDARY FEATURE ENHANCEMENT MODULE

Inspired by human perception, incorporating boundary information can significantly benefit segmentation and localization tasks involving glass recognition (He et al., 2021; Mei et al., 2023). To implement this concept, we introduce the Boundary Feature Enhancement (BFE) module, which is based on ASPP module (Chen et al., 2018; Mei et al., 2023), designed to identify and integrate boundary characteristics of glass into our transformer architecture. Contrary to the approach in Xie

et al. (2020), which uses an extra boundary stream (an encoder-decoder branch) for boundary feature extraction and integration with primary stream features, our BFE module is more streamlined. It derives boundary features directly from the targeted input features, bypassing the necessity for an additional stream.

As shown in Figure 1, the BFE module is designed to enhance feature learning before the last layer of our decoder so that the features can be subsequently improved by the reflection module in the next step. We empirically found that this placement of the BFE module has better performance and reduced memory usage compared to the placement of BFE at earlier layers of the decoder.

The BFE module then works as follows. The BFE begins by taking input features $\mathcal{X}_0$. These features are then processed through four parallel blocks, each dedicated to extracting multi-scale boundary features $\mathcal{F}_i(.)$ for $i = 1, 2, 3, 4$. Within each block, a convolution layer ($C(.)$) with different kernels and paddings is followed by batch normalization ($BN(.)$) and ReLU activation ($ReLU(.)$) operations, resulting in $\mathcal{F}i = ReLU(BN(C(X)))$. These multi-scale boundary features are subsequently fused using the Fusion module ($\mathcal{F}_{\text{fuse}} = C(\mathcal{F}_1 + \mathcal{F}_2 + \mathcal{F}_3 + \mathcal{F}_4)$), effectively aggregating shape properties and forming the glass boundary features. The output of the Fusion module then undergoes a convolutional layer to predict the boundary map, supervised by the Boundary loss. Finally, the enhanced boundary features $\mathcal{X}_e$ are obtained by aggregating the output of the Fusion module with the input features to precisely locate glass regions, especially their boundaries, as expressed by the following equation:

$$\mathcal{X}_e = (\mathcal{F}_{\text{fuse}}(\mathcal{F}_i(X)) + 1) \times \mathcal{X}_0 \tag{1}$$

where $+$ denotes element-wise addition, $\times$ denotes element-wise multiplication operations. Please see the appendix Figure 7 for the architecture of the BFE module.

**Boundary loss.** The Sobel kernel, sometimes called the Sobel-Feldman filter, is widely used in image processing and computer vision, used mainly for edge detection. It highlights image boundaries by analyzing the 2D gradient and emphasizing high spatial frequency regions. Our Boundary loss ($\mathcal{L}_b$) leverages the Sobel filter to measure how closely the gradients of a predicted mask match those of the ground truth mask, employing the Dice loss (Milletari et al., 2016):

$$\mathcal{L}_b = \text{dice}(\nabla_x \hat{M} \oplus \nabla_y \hat{M}, \nabla_x M_{GT} \oplus \nabla_y M_{GT}) \tag{2}$$

where $\hat{M}$ is the predicted object mask and $M_{GT}$ is the ground truth object mask. $\nabla_x$ and $\nabla_y$ denote the gradient along $x$-axis and $y$-axis computed by the Sobel filter. $\oplus$ represents the combination of the gradient maps into a single feature map. In our implementation, we define the combination $\oplus$ by:

$$a \oplus b = \max\left(\frac{1}{2}(a + b), \tau\right) \tag{3}$$

where $\tau$ is set to 0.01 to reduce noise in the gradient maps.

## 3.3 REFLECTION FEATURE ENHANCEMENT MODULE

To enhance the recognition of glass surfaces, we introduce the Reflection Feature Enhancement (RFE) module, capitalizing on the high reflectivity of glass when illuminated by light. These reflections provide valuable cues for recognizing glass surfaces in images (Yang et al., 2019; Mei et al., 2021). In our design, the RFE module is placed after the last layer of the decoder, after the boundary feature enhancement module. The RFE module employs a sophisticated convolution-deconvolution architecture (Yu & Koltun, 2016), which takes input features $Y$ and produces an enhanced feature map $Y_e$. This architecture allows the module to capture and process information at multiple levels of abstraction, which is essential for handling complex visual cues like reflections. Unlike the other reflection removal model (Zhang et al., 2018) that primarily addresses global reflections (assuming the entire input image is covered by glass), our RFE module targets detecting local reflections to locate glass surfaces. Please see the appendix Figure 7 for the architecture of the RFE module.

In detail, the encoder network $\mathcal{E}$ is responsible for extracting relevant features (the flow is highlighted by red arrows in Figure 7) from the input data. It consists of five blocks, each composed of a convolutional layer followed by batch normalization, ReLU activation, and either a Max-Pooling layer $\mathcal{P}_{max}(.)$ or an Upsampling layer $\mathcal{P}_{up}(.)$. Each encoder block can be defined as follows:

$$\mathcal{E}_i = \mathcal{P}^i\left(\text{ReLU}\left(\text{BN}\left(\text{C}(\mathcal{E}_{i-1})\right)\right)\right), \quad i \in [1..5] \tag{4}$$

where $\mathcal{E}_0 = C(Y)$, $\mathcal{P}^i$ is the Max-Pooling or Upsampling layer and when $i = 5$, $\mathcal{P}^i$ will be $\mathcal{P}_{Up}(.)$ instead of $\mathcal{P}_{max}(\cdot)$.

The decoder network $\mathcal{D}$ works in conjunction with the encoder to reconstruct and enhance the features (the flow is highlighted by blue arrows in Figure 7). It also comprises four blocks, interconnected by an Upsampling layer $\mathcal{P}_{up}(.)$ or a Deconvolutional layer $DC(\cdot)$, along with batch normalization and ReLU activation. Notably, the output of the preceding decoder block and the corresponding feature map $e_i = \mathcal{E}_i$ from the encoder block are concatenated before being fed into the subsequent decoder block. This facilitates the seamless flow of information across the network, enhancing its ability to capture and retain essential features of the reflective areas. Each decoder block can be formulated as follows:

$$\mathcal{D}_j = \mathcal{P}^j(\text{ReLU}(\text{BN}(\text{DC}(\mathcal{D}_{j-1} \otimes \mathcal{E}_{5-j})))), \quad j \in [1..4] \tag{5}$$

where $\mathcal{D}_0 = \mathcal{E}_5$, $\mathcal{P}^j$ is the Upsampling or Deconvolutional layer and when $j = 4$, $\mathcal{P}^j$ will be $DC(\cdot)$ instead of $\mathcal{P}_{Up}(\cdot)$.

The output of the decoder network is split into two tensors: the first tensor represents reflection mask $M_{\text{rf}}$, utilized for optimizing the reflection loss, while the second tensor contains the enhanced reflective features $Y_e$, which have been processed to capture and emphasize reflection-related information.

### 3.4 TRAINING

We use a softmax cross-entropy loss as our losses for supervising the semantic mask prediction and the ground truth semantic mask. Our loss for semantic mask prediction is

$$\mathcal{L}_s = \text{ce}(\hat{M}, M_{GT}) \tag{6}$$

where $\text{ce}(.)$ is the softmax cross-entropy loss.

Our loss for reflection mask prediction is as follows. Given that there is no ground truth for the reflection mask, we assume pseudo ground truth for the reflection mask to span common categories with reflective appearance such as window, door, cup, bottle, etc. The reflection loss is:

$$\mathcal{L}_r = \text{ce}(M_{\text{rf}}, \phi(M_{GT})) \tag{7}$$

where $\phi(.)$ is a function to extract pseudo ground truth with the reflective appearance in the ground truth semantic map $M_{GT}$. Note that as our pseudo ground truth might contain opaque appearances, we empirically found that such noise is not severe enough to affect the performance of the RFE module negatively.

The total loss for our training is:

$$\mathcal{L} = \alpha\mathcal{L}_s + \beta\mathcal{L}_b + \gamma\mathcal{L}_r \tag{8}$$

where $\alpha$, $\beta$ and $\gamma$ are hyper-parameters and are empirically set as [1.0,0.1,0.1] according to the experimental results.

## 4 EXPERIMENTS

To evaluate the proposed method, we carry out comprehensive experiments on glass (transparent) segmentation datasets (Trans10k-v2 (Xie et al., 2021b), RGBP-Glass (Mei et al., 2022), GSD-S (Lin et al., 2022)), mirror (reflection) segmentation datasets (MSD (Yang et al., 2019), PMD (Lin et al., 2020), RGBD-Mirror (Mei et al., 2021)), and generic segmentation (including both glass and mirror) datasets (TROSD (Sun et al., 2023), Stanford2D3D (Armeni et al., 2017)). For details of datasets, method implementations, experiments, and further analyses, please refer to the appendix.

### 4.1 QUALITATIVE AND QUANTITATIVE RESULTS

We evaluated the performance of our method across three distinct tasks: glass segmentation, mirror segmentation, and generic segmentation. The experimental results show that our method consistently outperforms other state-of-the-art (SOTA) methods across all datasets. To clarify, we create several

TransCues models (Ours-X with X is postfixes: -T, -S, -M, -L, -B1, -B2, -B3, -B4, and -B5, represented the size of the model as PVTv1 Tiny, Small, Medium, Large, and PVTv2 B1-5, respectively) at different scales for discussion in the following sections.

**Glass Segmentation.** We benchmarked our method against recent glass segmentation methods on binary and semantic segmentation tasks. For the binary glass segmentation task, as shown in Table 1, our method (Ours-B4) achieves the pinnacle of mIoU(%) scores, outpacing all other competing methods, which include (SegFormer (Xie et al., 2021a), GSD (Lin et al., 2021), SETR (Zheng et al., 2021), GDNet (Mei et al., 2020)). Specifically, it surpasses the runner-up method by margins of 4.35% for RGB-P and 2% for GSD-S. Noteworthy is our method's balance of performance with computational efficiency, registering relatively lower GFLOPs compared to its peers. Shifting our focus to the semantic glass segmentation task, where the challenge extends beyond merely detecting glass areas to classifying them into 11 fine-grained categories, our method still reigns supreme. It surpasses competing approaches such as (Trans4Trans (Zhang et al., 2022), DenseASPP (Yang et al., 2018), DeepLabv3+ (Chen et al., 2018), OCNet (Yuan et al., 2021b), Trans2Seg (Xie et al., 2021b)) by a substantial 4.15% margin in terms of mIoU performance. This dominance in accuracy does not come at the expense of efficiency, as evidenced in Table 2. These comprehensive evaluations underscore the effectiveness of our approach across diverse glass segmentation scenarios, affirming its position as a top-performing and computationally efficient choice for these tasks.

As shown in Figure 3, we can observe that the recent approaches, such as GDNet and Trans2Seg, may over-detect glass regions in specific images but under-detect glass regions like GSD. In contrast, our method can accurately identify glass portions of diverse dimensions and morphologies, effectively differentiating them from look-alike non-glass regions in complex images (such as the one showcased in the top right), thanks to the BFE and RFE modules, which leverage boundary and reflection cues, help our method perform better in challenging scenarios.

| Method | Backbone | GFLOPs ↓ | RGB-P | GSD-S |
|---|---|---|---|---|
| SegFormer | MiT-B5 | 70.2 | 78.4 | 54.7 |
| Ours-B4 | PVTv2-B4 | 79.3 | **82.1** | **74.1** |
| GSD | ResNeXt-101 | 92.7 | 78.1 | 72.1 |
| SETR | ViT-Large | 240.1 | 77.6 | 56.7 |
| GDNet | ResNeXt-101 | 271.5 | 77.6 | 52.9 |

Table 1: Binary Glass Segmentation on RGB-P, GSD-S. We reported mIoU(%) for both datasets.

| Method | GFLOPs ↓ | Accuracy ↑ | mIoU ↑ |
|---|---|---|---|
| Trans4Trans-M | 34.38 | 95.01 | 75.14 |
| DenseASPP | 36.20 | 90.86 | 63.01 |
| Ours-B2 | 37.03 | **95.92** | **79.29** |
| DeepLabv3+ | 37.98 | 92.75 | 68.87 |
| OCNet | 43.31 | 92.03 | 66.31 |
| Trans2Seg | 49.03 | 94.14 | 72.15 |

Table 2: Semantic Glass Segmentation on Trans10K-v2.

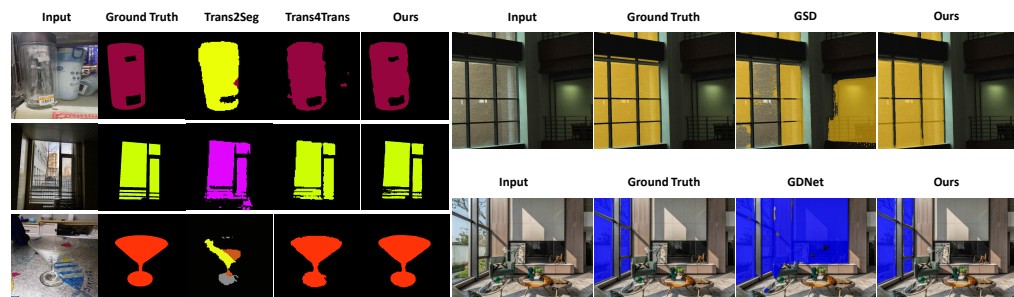

Figure 3: Comparison of glass segmentation methods on Trans10K-v2 (left), RGB-P (top-right), and GSD-S (bottom-right) datasets.

**Mirror Segmentation.** To demonstrate the robustness of our approach when dealing with reflective surfaces, we rigorously evaluated our method on three standard binary mirror segmentation datasets: MSD, PMD, and RGBD-Mirror. These datasets are chosen for their resemblance to the reflective characteristics inherent in glass objects. Ours-B3 was chosen as a representative model for a balanced and comparable evaluation. We compared its performance against recent state-of-the-art methodologies, such as SANet (Guan et al., 2022), VCNet (Tan et al., 2023), and SATNet (Huang et al., 2023). The outcomes, detailed in Table 3, unequivocally highlight our method's supremacy in mirror segmentation, surpassing the competition across various metrics.

| Method | Backbone | MSD | | | PMD | | | RGBD-M | | |
|--------|----------|-----|-----|------|-----|-----|------|--------|-----|------|
| | | IoU ↑ | $F_\beta$ ↑ | MAE ↓ | IoU ↑ | $F_\beta$ ↑ | MAE ↓ | IoU ↑ | $F_\beta$ ↑ | MAE ↓ |
| SANet | ResNeXt101 | 79.85 | 0.879 | 0.054 | 66.84 | 0.837 | 0.032 | 74.99 | 0.873 | 0.048 |
| VCNet | ResNeXt101 | 80.08 | 0.898 | 0.044 | 64.02 | 0.815 | 0.028 | 73.01 | 0.849 | 0.052 |
| SATNet | Swin-S | 85.41 | 0.922 | 0.033 | 69.38 | 0.847 | 0.025 | 78.42 | 0.906 | 0.031 |
| Ours-B3 | PVTv2-B3 | **91.04** | **0.953** | **0.028** | **69.61** | **0.853** | **0.021** | **88.52** | **0.954** | **0.027** |

Table 3: Binary Mirror Segmentation on MSD, PMD, and RGBD-Mirror datasets. Our method achieves the best performance in terms of all the evaluation metrics.

**Generic Segmentation.** In addition to evaluating our method, we compared with existing approaches (Trans4Trans (Zhang et al., 2022), TransLab (Xie et al., 2020), DANet (Fu et al., 2019), TROSNet (Sun et al., 2023), Trans2Seg (Xie et al., 2021b), PVT (Wang et al., 2021b)). As detailed in Table 4, our method outshines SOTAs competitors on the TROSD dataset (a dedicated dataset for transparent and reflective object segmentation), underscoring its effectiveness in handling complex transparent and reflective object segmentation. This is mainly because our approach focuses on low-level features, making it possible to accurately identify and preserve content differences along the borders of transparent and reflective objects. Moreover, to show our generalization ability, we test our method on the large-scale real-world Stanford2D3D dataset consisting of common and transparent objects (less than 1% of the total images). As shown in Table 5, our method outperforms other existing works (about 8.25% better performance in mIOU) in semantic scene segmentation, demonstrating its robustness in discerning specific objects' appearance and normal scenes.

| Method | IOU ↑ | | | mIoU ↑ | mAcc ↑ |
|--------|-------|-------|-------|--------|--------|
| | R | T | B | | |
| TransLab | 42.57 | 50.72 | 96.01 | 63.11 | 68.72 |
| DANet | 42.76 | 54.39 | 95.88 | 64.34 | 70.95 |
| TROSNet | 48.75 | 48.56 | 95.49 | 64.26 | 75.93 |
| Ours-B3 | **67.25** | **67.23** | **97.69** | **77.39** | **87.62** |

Table 4: Comparison of various methods on TROSD dataset. R: reflective objects. T: transparent objects. B: background.

| Method | GFLOPs ↓ | MParams ↓ | mIoU ↑ |
|--------|----------|-----------|--------|
| Tran4Tran-M | 34.38 | 43.65 | 45.73 |
| Ours-M | 34.51 | 43.70 | 52.57 |
| Ours-B2 | 37.03 | 27.59 | **53.98** |
| Trans2Seg-M | 40.98 | 30.53 | 43.83 |
| PVT-M | 49.00 | 56.20 | 42.49 |

Table 5: Comparison with state-of-the-art methods on Stanford2D3D dataset.

**Failure Cases.** Figure 4-left showcases failure cases of our method and other methods on Trans10K-v2. Our method would confuse and fail to segment the object with similar properties as others. In such a scenario, even human beings would struggle to differentiate between these transparent things. However, despite assigning the wrong label, our method can still maintain the object's shape. We also show several failure instances (Figure 4-right) in our system that misinterpret non-glass areas as glass because they seem and behave the same, such as still in the door frame with reflection and distortion.

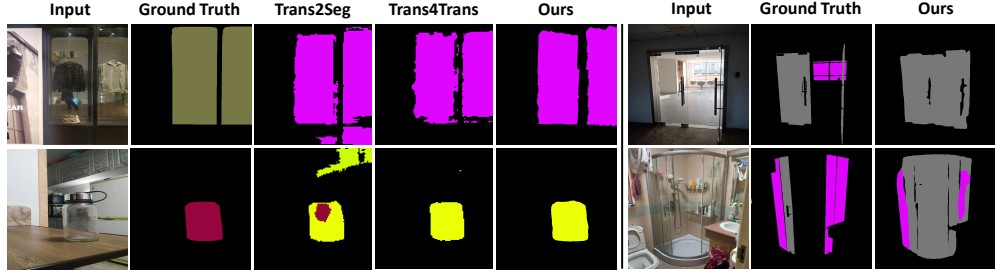

Figure 4: Some failure cases of our method and existing methods on the Trans10K-v2 dataset.

## 4.2 ABLATION STUDIES

In this section, we present ablation studies to verify different aspects of the design of our model and underscore the significance of each module within our model. Any alterations or omissions to the proposed design led to noticeable drops in performance, which justifies our choice of transformer architecture and the boundary and reflection feature learning components.

**Analysis of backbone architectures.** We conducted experiments on various backbones, as presented in Figure 5. Among these options, the PVT-v2 backbone (Wang et al., 2022) stands out with significantly higher mIoU and remarkably compact model size (MParams). Despite its higher complexity (GFLOPs) compared to the FocalNet backbone (Yang et al., 2022), it still manages to achieve better performance. Additionally, the PVT-v2 backbone demonstrates a lower complexity than the DaViT backbone (Ding et al., 2022) while maintaining competitive results. These findings highlight the superiority of the PVT-v2 backbone in achieving an optimal balance between performance and model size, making it a promising choice for our method. When comparing PVT-v1 (Wang et al., 2021b) with other backbones, it boasts a considerably smaller model size and lower complexity. Despite these improvements, its performance is comparable to other backbones. The PVT-v1 backbone is efficient since it performs similarly while being lighter and less computationally intensive.

**Effectiveness of different modules.** To assess the contribution of both the BFE and RFE modules to our architecture's performance, we systematically evaluated the model under various configurations: **(1) Baseline Model** (PVTv1-T or PVTv2-B1 without BFE and RFE): this served as our control group, where both the BFE and RFE modules were excluded. Results indicate a foundational performance that the other configurations could be compared against. **(2) Incorporation of BFE:** when only the BFE module was integrated into our network, we noticed a significant performance enhancement. However, this performance did not reach the potential of the combined BFE and RFE configuration. This proved that while BFE is essential, it works best in tandem with RFE. **(3) Incorporation of RFE:** similarly, adding only the RFE module to the baseline network also provided an uptick in performance. This emphasized the value of detecting reflections in transparent objects for the segmentation task. **(4) Combined Integration of BFE and RFE:** both modules were simultaneously integrated into our network. The performance gain observed in this configuration, as shown in Table 6, was the most pronounced, with gains of **6.36%** and **7.61%** in mIoU on the Trans10K-v2 and Stanford2D3D datasets, respectively. This confirms that the combined effects of boundary and reflection cues significantly augment the network's segmentation capabilities.

Interestingly, the ablation studies further provide an explanation of why our method works well for generic segmentation as in the Stanford2D3D dataset. It can be seen in Table 6 that the boundary module yields the most significant performance gain compared to the reflection module. This means that for generic segmentation, where reflection feature learning has negligible improvement, our boundary feature learning remains effective for general semantic labels.

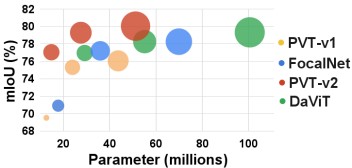

| Backbone | FLOPs | Params | BFE | RFE | S2D3D | Trans10K |
|----------|-------|--------|-----|-----|-------|----------|
| PVTv1-T | 10.16 | 13.11 | - | - | 45.19 | 69.44 |
| PVTv2-B1 | 11.48 | 13.89 | - | - | 46.79 +1.6 | 70.49 +1.05 |
| PVTv2-B1 | 13.22 | 14.37 | - | ✓ | 48.12 +2.93 | 72.65 +3.21 |
| PVTv2-B1 | 19.55 | 14.39 | ✓ | - | 50.22 +5.03 | 74.89 +5.45 |
| PVTv2-B1 | 21.29 | 14.87 | ✓ | ✓ | **51.55** +6.36 | **77.05** +7.61 |

Figure 5: Different backbones. The bubble's size is its speed in GFLOPs.

Table 6: Effectiveness of different modules of our method. The last row corresponds to our method (Ours-B1). The metric is mIoU.

## 5 CONCLUSION

This paper proposes a method to segment transparent and opaque objects along with general objects using pyramidal transformer architecture. Our method exploits two important visual cues based on boundary and reflection features that significantly lead to performance gain in both transparent and generic segmentation tasks, respectively. We extensively evaluated our proposed method on several benchmark datasets, demonstrating the robustness of our method in various scenarios.

Our architecture is a fully transformer-based method built upon the PVT. Therefore, some limitations still exist that lower our method's capabilities for visual tasks. Firstly, the position encoding of our network is fixed-size, requiring a resizing step, which will damage and distort the object's shape. Secondly, similar to other vision transformer-based methods, the computational cost of our network is relatively high when dealing with high-resolution images. Finally, as stated before, we use the same position embedding as ViT and PVT, which is insufficient for arbitrary resolution of input images. In future work, we would like to investigate the extension of our method to other modalities, including depth images, event data, videos, and dynamic scenes.

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

Our appendix has four sections. Appendix A shows the detailed architecture of each module in our proposed method. Appendix B provides hyper-parameters of each scale of PVTv1 and PVTv2. Appendix C contains experimental setup such as datasets, implementation details, and evaluation metrics. Additional analysis is detailed in Appendix D, together with quantitative and qualitative results of each dataset.

# A NETWORK ARCHITECTURE

In this section, we show the detailed architecture of the Feature Extraction Module in our encoder and the Feature Parsing Module in our decoder in Figure 6, and then the Boundary Feature Enhancement and Reflection Feature Enhancement modules in Figure 7.

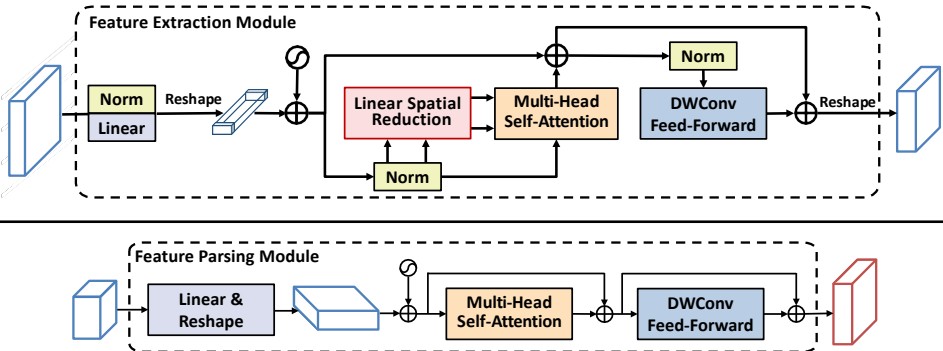

Figure 6: The architecture of the Feature Extraction Module (top) in our encoder and Feature Parsing Module (bottom) in our decoder. Zoom in for better visualization.

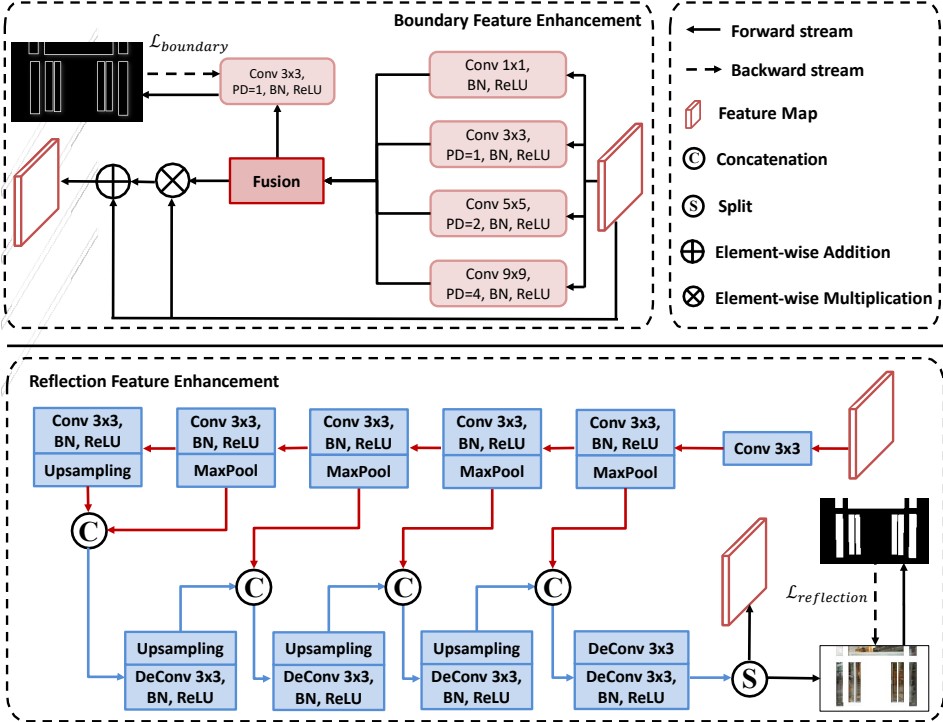

Figure 7: The architecture of Boundary Feature Enhancement (top) and Reflection Feature Enhancement (bottom). Zoom in for better visualization.

# B  BACKBONE SETTINGS

The hyper-parameters of backbones in our models are listed as follows:

- $S_i$: stride of overlapping patch embedding in Stage $i$;
- $C_i$: channel number of output of Stage $i$;
- $L_i$: number of encoder layers in Stage $i$;
- $R_i$: reduction ratio of SRA layer in Stage $i$;
- $P_i$: patch size of Stage $i$;
- $N_i$: head number of Efficient Self-Attention in Stage $i$;
- $E_i$: expansion ratio of Feed-Forward layer (Vaswani et al., 2017) in Stage $i$;

In addition, we describe a series of PVTv1 (Wang et al., 2021b) backbones with different scales (Tiny, Small, Medium, and Large) in Table 7 and a series of PVTv2 (Wang et al., 2022) backbones with different scales (B1 to B5) in Table 8.

| | Output Size | Layer Name | PVT-Tiny | | PVT-Small | | PVT-Medium | | PVT-Large | |
|---|---|---|---|---|---|---|---|---|---|---|
| **Stage 1** | $\frac{H}{4} \times \frac{W}{4}$ | Patch Embedding | $P_1 = 4;\ C_1 = 64$ | | | | | | | |
| | | Transformer Encoder | $R_1 = 8$ $N_1 = 1$ $E_1 = 8$ | $\times 2$ | $R_1 = 8$ $N_1 = 1$ $E_1 = 8$ | $\times 3$ | $R_1 = 8$ $N_1 = 1$ $E_1 = 8$ | $\times 3$ | $R_1 = 8$ $N_1 = 1$ $E_1 = 8$ | $\times 3$ |
| **Stage 2** | $\frac{H}{8} \times \frac{W}{8}$ | Patch Embedding | $P_2 = 2;\ C_2 = 128$ | | | | | | | |
| | | Transformer Encoder | $R_2 = 4$ $N_2 = 2$ $E_2 = 8$ | $\times 2$ | $R_2 = 4$ $N_2 = 2$ $E_2 = 8$ | $\times 3$ | $R_2 = 4$ $N_2 = 2$ $E_2 = 8$ | $\times 3$ | $R_2 = 4$ $N_2 = 2$ $E_2 = 8$ | $\times 8$ |
| **Stage 3** | $\frac{H}{16} \times \frac{W}{16}$ | Patch Embedding | $P_3 = 2;\ C_3 = 320$ | | | | | | | |
| | | Transformer Encoder | $R_3 = 2$ $N_3 = 5$ $E_3 = 4$ | $\times 2$ | $R_3 = 2$ $N_3 = 5$ $E_3 = 4$ | $\times 6$ | $R_3 = 2$ $N_3 = 5$ $E_3 = 4$ | $\times 18$ | $R_3 = 2$ $N_3 = 5$ $E_3 = 4$ | $\times 27$ |
| **Stage 4** | $\frac{H}{32} \times \frac{W}{32}$ | Patch Embedding | $P_4 = 2;\ C_4 = 512$ | | | | | | | |
| | | Transformer Encoder | $R_4 = 1$ $N_4 = 8$ $E_4 = 4$ | $\times 2$ | $R_4 = 1$ $N_4 = 8$ $E_4 = 4$ | $\times 3$ | $R_4 = 1$ $N_4 = 8$ $E_4 = 4$ | $\times 3$ | $R_4 = 1$ $N_4 = 8$ $E_4 = 4$ | $\times 3$ |

Table 7: Detailed settings of PVTv1 series which is adopted from (Wang et al., 2021b).

# C  EXPERIMENTAL SETUP

## C.1  DATASETS

We comprehensively evaluated our proposed method on diverse datasets to demonstrate its exceptional performance and versatility. These datasets encompass a broad spectrum of segmentation tasks such as Glass (Transparent) datasets (Trans10k-v2 (Xie et al., 2021b), RGBP-Glass (Mei et al., 2022), and GSD-S (Lin et al., 2022)), Mirror (Reflection) datasets (MSD (Yang et al., 2019), PMD (Lin et al., 2020), and RGBD-Mirror (Mei et al., 2021)), and generic datasets, which consists of both glass and mirror objects (TROSD (Sun et al., 2023), and Stanford2D3D (Armeni et al., 2017)), ranging from binary to semantic segmentation, with a particular focus on images featuring reflective, transparent, or both characteristics. Our evaluation also considers the varied positions and fields of view (FOV) of objects within the images. Objects of interest may appear near or far from the camera's perspective, positioned randomly or at the center of the frame, providing a rich and realistic testing environment. Furthermore, the datasets we utilized are substantial in size, ensuring coverage of a broad range of environmental and scenario complexities. This encompasses indoor and outdoor scenarios, as well as varying lighting conditions, diverse object scales, different viewpoints, and levels of occlusion. Our extensive evaluation showcases the robustness and adaptability of our method across a wide array of real-world conditions. The detail of each dataset is shown in Table 9.

| | Output Size | Layer Name | PVT-B1 | PVT-B2 | PVT-B3 | PVT-B4 | PVT-B5 |
|---|---|---|---|---|---|---|---|
| **Stage 1** | $\frac{H}{4} \times \frac{W}{4}$ | Overlapping Patch Embedding | \multicolumn{5}{c}{$S_1 = 4$ ; $C_1 = 64$} | | | | |
| | | Transformer Encoder | $R_1=8$ $N_1=1$ $E_1=8$ $L_1=2$ | $R_1=8$ $N_1=1$ $E_1=8$ $L_1=3$ | $R_1=8$ $N_1=1$ $E_1=8$ $L_1=3$ | $R_1=8$ $N_1=1$ $E_1=8$ $L_1=3$ | $R_1=8$ $N_1=1$ $E_1=4$ $L_1=3$ |
| **Stage 2** | $\frac{H}{8} \times \frac{W}{8}$ | Overlapping Patch Embedding | \multicolumn{5}{c}{$S_2 = 2$ ; $C_2 = 128$} | | | | |
| | | Transformer Encoder | $R_2=4$ $N_2=2$ $E_2=8$ $L_2=2$ | $R_2=4$ $N_2=2$ $E_2=8$ $L_2=3$ | $R_2=4$ $N_2=2$ $E_2=8$ $L_2=3$ | $R_2=4$ $N_2=2$ $E_2=8$ $L_2=8$ | $R_2=4$ $N_2=2$ $E_2=4$ $L_2=6$ |
| **Stage 3** | $\frac{H}{16} \times \frac{W}{16}$ | Overlapping Patch Embedding | \multicolumn{5}{c}{$S_3 = 2$ ; $C_3 = 320$} | | | | |
| | | Transformer Encoder | $R_3=2$ $N_3=5$ $E_3=4$ $L_3=2$ | $R_3=2$ $N_3=5$ $E_3=4$ $L_3=6$ | $R_3=2$ $N_3=5$ $E_3=4$ $L_3=18$ | $R_3=2$ $N_3=5$ $E_3=4$ $L_3=27$ | $R_3=2$ $N_3=5$ $E_3=4$ $L_3=40$ |
| **Stage 4** | $\frac{H}{32} \times \frac{W}{32}$ | Overlapping Patch Embedding | \multicolumn{5}{c}{$S_4 = 2$ ; $C_4 = 512$} | | | | |
| | | Transformer Encoder | $R_4=1$ $N_4=8$ $E_4=4$ $L_4=2$ | $R_4=1$ $N_4=8$ $E_4=4$ $L_4=3$ | $R_4=1$ $N_4=8$ $E_4=4$ $L_4=3$ | $R_4=1$ $N_4=8$ $E_4=4$ $L_4=3$ | $R_4=1$ $N_4=8$ $E_4=4$ $L_4=3$ |

Table 8: Detailed settings of PVTv2 series which is adopted from (Wang et al., 2022).

| Dataset | Modalities | Images | Tasks | Types | FOV | Position |
|---|---|---|---|---|---|---|
| Trans10k-v2 (Xie et al., 2021b) | RGB | 10,428 | semantic | both | both | random |
| RGBP-Glass (Mei et al., 2022) | RGB-P | 4,511 | binary | transparent | far | random |
| GSD-S (Lin et al., 2022) | RGB-S | 3,009 | binary | transparent | far | random |
| MSD (Yang et al., 2019) | RGB | 4,018 | binary | reflective | both | center |
| PMD (Lin et al., 2020) | RGB | 6,461 | binary | reflective | both | random |
| RGBD-Mirror (Mei et al., 2021) | RGB-D | 3,049 | binary | reflective | both | center |
| TROSD (Sun et al., 2023) | RGB-D | 11,060 | semantic | both | near | center |
| Stanford2D3D (Armeni et al., 2017) | RGB-D | 70,496 | semantic | both | far | random |

Table 9: Comparison between different datasets in our experiments. 'P,' 'D,' and 'S' means polarization images, depth images, and semantic map, respectively.

## C.2 IMPLEMENTATION DETAILS.

We implemented our method in PyTorch 1.8.0 and CUDA 11.2. We adopted AdamW optimizer (Loshchilov & Hutter, 2019) where the learning rate $\gamma$ was set to $10^{-4}$ with epsilon $10^{-8}$ and weight decay $10^{-4}$. Our model was trained with a batch size of 8 and on a single NVIDIA RTX 3090 GPU, but it still can be trained on an older 2080 Ti or 1080 Ti GPU with a smaller batch size, e.g., 4. We evaluated all the variants of our network on the validation set for every epoch during the training process. We used the best model of each variant on the validation set to evaluate the variant on the test set. The training process was completed once there were no further improvements achieved.

## C.3 EVALUATION METRICS

For our evaluation, we adopt four widely used metrics from (Mei et al., 2022) for quantitatively assessing the glass segmentation performance: mean intersection over union (mIoU), weighted F-measure ($F_\beta^w$), mean absolute error (MAE), and balance error rate (BER).

**Intersection over Union** $(IoU)$ is a widely used metric in segmentation tasks, which is defined as:

$$IoU = \frac{\sum\limits_{i=1}^{H} \sum\limits_{j=1}^{W} (G(i,j) * P_b(i,j))}{\sum\limits_{i=1}^{H} \sum\limits_{j=1}^{W} (G(i,j) + P_b(i,j) - G(i,j) * P_b(i,j))} \tag{9}$$

where $G$ is the ground truth mask in which the values of the glass region are 1 while those of the non-glass region are 0; $P_b$ is the predicted mask binarized with a threshold of 0.5; and $H$ and $W$ are the height and width of the ground truth mask, respectively.

**Weighted F-measure** $(F_\beta^w)$ is adopted from the salient object detection tasks with $\beta = 0.3$. F-measure $(F_\beta)$ measures the prediction map's precision and recall. Recent studies (Fan et al., 2017) have suggested that the weighted F-measure $(F_\beta^w)$ (Margolin et al., 2014) can provide more reliable evaluation results than the traditional $F_\beta$. Thus, we report $F_\beta^w$ in the comparison.

**Mean Absolute Error** (MAE) is widely used in foreground-background segmentation tasks, which calculates the element-wise difference between the prediction map $P$ and the ground truth mask $G$:

$$MAE = \frac{1}{H \times W} \sum\limits_{i=1}^{H} \sum\limits_{j=1}^{W} |P(i,j) - G(i,j)|, \tag{10}$$

where $P(i,j)$ indicates the predicted probability score at location $(i,j)$.

**Balance Error Rate** (BER) is a standard metric used in shadow detection tasks, defined as:

$$BER = (1 - \frac{1}{2}(\frac{TP}{N_p} + \frac{TN}{N_n})) \times 100 \tag{11}$$

where $TP, TN, N_p$, and $N_n$ represent the numbers of true positive pixels, true negative pixels, glass pixels, and non-glass pixels, respectively.

# D    EXPERIMENT RESULTS

## D.1    ADDITIONAL ANALYSIS

### D.1.1    DIFFERENT COMBINATIONS OF NETWORK ARCHITECTURE.

Table 10 presents the comparisons between various combinations of encoders and decoders, such as using only CNN architecture, using a variety of both CNN and Transformer, and using a fully transformer-based model. Our method, an encoder-decoder transformer-based model, outperforms other competitive networks, indicating the system's capability for effectively segmenting transparent objects. In this ablation study, we used Ours-M and Ours-B2 (not the best model Ours-B5), which has the same network size as other methods (-M model size), for a fair comparison.

| Method | Encoder | | Decoder | | GFLOPs | mIoU |
| --- | --- | --- | --- | --- | --- | --- |
| | **Transformer** | **CNN** | **Transformer** | **CNN** | | |
| Trans4Trans-M (Zhang et al., 2022) | ✓ | | ✓ | | 34.3 | 75.1 |
| Ours-M | ✓ | | ✓ | | 34.5 | 76.1 |
| Ours-B2 | ✓ | | ✓ | | 37.0 | **79.3** |
| Trans2Seg-M (Xie et al., 2021b) | | ✓ | ✓ | | 40.9 | 69.2 |
| FCN (Long et al., 2015) | | ✓ | | ✓ | 42.2 | 62.7 |
| OCNet (Yuan et al., 2021b) | | ✓ | | ✓ | 43.3 | 66.3 |
| PVT-M (Wang et al., 2021b) | ✓ | | ✓ | | 49.0 | 72.1 |

Table 10: Effectiveness of different network architecture combinations. Models are evaluated on the Trans10K-v2 dataset (Xie et al., 2021b). **Note that:** the best result is **bold**, the second best result is underline and the results are sorting by ascending of GFLOPS.

**Effectiveness of the embedding channel.** We experiment with the embedding channel with various values (64, 128, 256, 512) and report the mIoU and Accuracy of Ours-B1 model in Table 11 and

Figure 8. Throughout the results, we proved that our model achieved better performance with a higher embedding channel (from 77.05% at 64 channels to 78.85% at 512 channels). Note that, due to memory limits, we can not perform experiments with higher embedding channels, e.g., 1024, 2048, and to save computational resources, we used Ours-B1 in this ablation study.

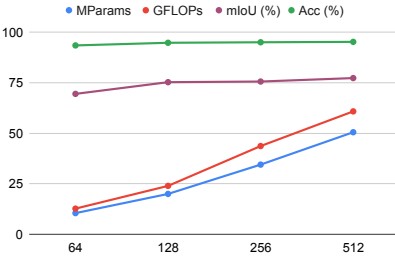

Figure 8: Plot of embedding channels.

| Channel | GFLOPs | MParams | Acc (%) | mIoU (%) |
|---|---|---|---|---|
| 64 | 21.99 | 14.87 | 95.37 | 77.05 |
| 128 | 40.01 | 25.78 | 95.83 | 77.82 |
| 256 | 64.51 | 44.92 | 96.08 | 78.24 |
| 512 | 100.54 | 61.68 | 96.38 | 78.85 |

Table 11: Effectiveness of the embedding channel in our Transformer architecture on Trans10K-v2 (Xie et al., 2021b).

**Real-time performance.** We calculate the inference speed of our models on different GPUs (NVIDIA GTX 1070, NVIDIA RTX 3090) with the resolution of $512 \times 512$ and batch size of 1. As shown in Table 12, While Our-T model exhibits a notably lower computational cost than the other versions, it's important to note that all these models deliver performance levels well-suited for deployment on robotic systems. In real-world situations, reaching a similar level of prediction accuracy on each frame is crucial because it makes it possible for a navigation system to be more responsive, improving the system's capacity to aid robots efficiently.

| Network | NVIDIA 1070 (ms) ↓ | NVIDIA 3090 (ms) ↓ |
|---|---|---|
| Ours-T | 15.8 ±0.9 | 10.8 ±0.4 |
| Ours-S | 26.6 ±0.8 | 14.1 ±0.5 |
| Ours-B1 | 32.9 ±1.3 | 16.7 ±0.3 |
| Ours-B2 | 46.7 ±1.1 | 19.4 ±0.7 |

Table 12: Inference time (ms/frame) of our methods are tested on various GPUs at $512 \times 512$ on Trans10K-v2 (Xie et al., 2021b).

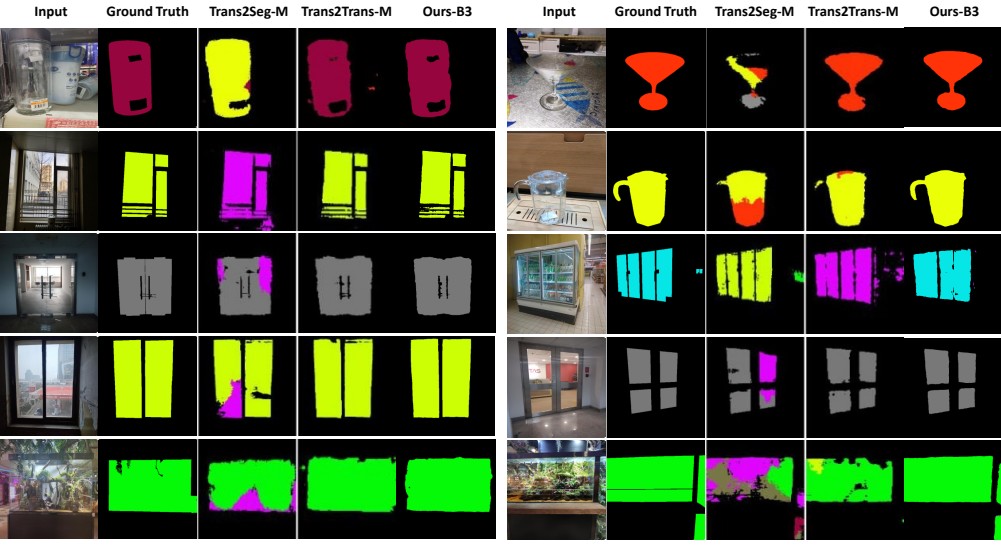

Figure 9: Qualitative valuation of our method and existing methods on Trans10K-v2 (Xie et al., 2021b). Note that we used Ours-B3, which has the same network size as other methods (-M model), for a fair comparison.

### D.2 COMPARISON ON GLASS OBJECT SEGMENTATION

**Trans10k-v2 dataset.** We provided a detailed comparison with category IoU of all classes in Trans10k-v2 dataset (Xie et al., 2021b) of our method and SOTAs in Table 13, and more visualizations in Figure 9. Compared with existing methods, our fish tanks are accurately obtained. In addition, the results also show that our method is better than existing works in the case that the scene consists of reflection; our method can still segment the area of the glass area.

**RGBP-Glass dataset.** We extensively compare the effectiveness of our method with state-of-the-art methods, as shown in Table 14. All methods are retrained on RGBP-Glass dataset (Mei et al., 2022) for a fair comparison. EAFNet (Xiang et al., 2021), Polarized Mask R-CNN (P.M. R-CNN) (Kalra et al., 2020), and PGSNet (Mei et al., 2022) are the three methods that leverage polarization cues. SETR (Zheng et al., 2021), SegFormer (Xie et al., 2021a) are the two methods focusing on general

| Method | GFLOPs ↓ | ACC ↑ | mIoU ↑ |
|---|---|---|---|
| FPENet (Liu & Yin, 2019) | 0.76 | 70.31 | 10.14 |
| ESPNetv2 (Mehta et al., 2019) | 0.83 | 73.03 | 12.27 |
| ContextNet (Poudel et al., 2018) | 0.87 | 86.75 | 46.69 |
| FastSCNN (Poudel et al., 2019) | 1.01 | 88.05 | 51.93 |
| DFANet (Li et al., 2019b) | 1.02 | 85.15 | 42.54 |
| ENet (Paszke et al., 2016) | 2.09 | 71.67 | 8.50 |
| DeepLabv3+MBv2 (Sandler et al., 2018) | 2.62 | 88.39 | 54.16 |
| HRNet_w18 (Wang et al., 2020) | 4.20 | 89.58 | 54.25 |
| HarDNet (Chao et al., 2019) | 4.42 | 90.19 | 56.19 |
| DABNet (Li et al., 2019a) | 5.18 | 77.43 | 15.27 |
| LEDNet (Wang et al., 2019) | 6.23 | 86.07 | 46.40 |
| Trans4Trans-T (Zhang et al., 2022) | 10.45 | 93.23 | 68.63 |
| Ours-T | 10.50 | **93.52** | **69.53** |
| ICNet (Zhao et al., 2018a) | 10.64 | 78.23 | 23.39 |
| BiSeNet (Yu et al., 2018) | 19.91 | 89.13 | 58.40 |
| Trans4Trans-S (Zhang et al., 2022) | 19.92 | 94.57 | 74.15 |
| Ours-S | 20.00 | 94.83 | 75.32 |
| Ours-B1 | 21.29 | **95.37** | **77.05** |
| Trans4Trans-M (Zhang et al., 2022) | 34.38 | 95.01 | 75.14 |
| Ours-M | 34.51 | 95.08 | 76.06 |
| DenseASPP (Yang et al., 2018) | 36.20 | 90.86 | 63.01 |
| Ours-B2 | 37.03 | **95.92** | **79.29** |
| DeepLabv3+ (Chen et al., 2018) | 37.98 | 92.75 | 68.87 |
| FCN (Long et al., 2015) | 42.23 | 91.65 | 62.75 |
| OCNet (Yuan et al., 2021b) | 43.31 | 92.03 | 66.31 |
| RefineNet (Lin et al., 2017) | 44.56 | 87.99 | 58.18 |
| Trans2Seg (Xie et al., 2021b) | 49.03 | 94.14 | 72.15 |
| Ours-L | 50.54 | 95.28 | 77.35 |
| TransLab (Xie et al., 2020) | 61.31 | 92.67 | 69.00 |
| Ours-B3 | 68.35 | 96.28 | 80.04 |
| Ours-B4 | 79.34 | **96.59** | **80.99** |
| U-Net (Ronneberger et al., 2015) | 124.55 | 81.90 | 29.23 |
| DUNet (Jin et al., 2019) | 123.69 | 90.67 | 59.01 |
| Ours-B5 | 154.37 | **96.93** | **81.37** |
| DANet (Fu et al., 2019) | 198.00 | 92.70 | 68.81 |
| PSPNet (Zhao et al., 2017) | 187.03 | 92.47 | 68.23 |

Table 13: Quantitative evaluation of our method and existing methods on the Trans10K-v2 dataset (Xie et al., 2021b). **Note that:** the best result is **bold**, the second best result is underline and the results are sorting by ascending of GFLOPS.

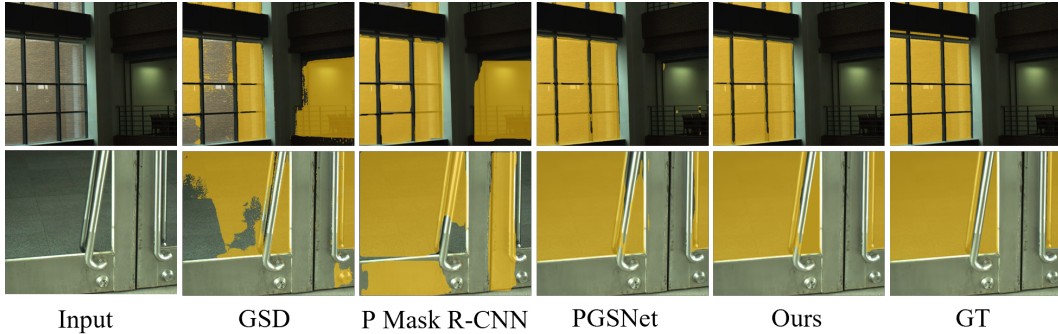

| Input | GSD | P Mask R-CNN | PGSNet | Ours | GT |

Figure 10: Qualitative comparison of our method with other methods on RGB-P dataset (Mei et al., 2022).

| Method | Backbone | GFLOPs↓ | mIoU↑ | $F_\beta^w$↑ | MAE↓ | BER↓ |
|---|---|---|---|---|---|---|
| EAFNet (Xiang et al., 2021) ∗ | ResNet-18 | 18.93 | 53.86 | 0.611 | 0.237 | 24.65 |
| P.M. R-CNN (Kalra et al., 2020) ∗ | ResNet-101 | 56.59 | 66.03 | 0.714 | 0.178 | 18.92 |
| PGSNet (Mei et al., 2022) ∗ | Conformer-B | 290.62 | 81.08 | 0.842 | 0.091 | 9.63 |
| Trans2Seg (Xie et al., 2021b) | ResNet-50 | 49.03 | 75.21 | 0.799 | 0.122 | 13.23 |
| TransLab (Xie et al., 2020) | ResNet-50 | 61.26 | 73.59 | 0.772 | 0.148 | 15.73 |
| SegFormer (Xie et al., 2021a) | MiT-B5 | 70.24 | 78.42 | 0.815 | 0.121 | 13.03 |
| GSD (Lin et al., 2021) | ResNeXt-101 | 92.69 | 78.11 | 0.806 | 0.122 | 12.61 |
| Ours-B5 | PVTv2-B5 | 154.37 | **82.77** | **0.879** | **0.042** | **9.59** |
| GDNet (Mei et al., 2020) | ResNeXt-101 | 271.53 | 77.64 | 0.807 | 0.119 | 11.79 |
| SETR (Zheng et al., 2021) | ViT-Large | 240.11 | 77.60 | 0.817 | 0.114 | 11.46 |

Table 14: Quantitative comparison against state-of-the-art on RGB-P dataset (Mei et al., 2022). **Note that:** we only use RGB as input for our method, and the results are sorted by ascending of GFLOPS. (∗) denotes the glass segmentation methods with additional polarization images as input.

semantic/instance segmentation tasks. GDNet (Mei et al., 2020), TransLab (Xie et al., 2020), Trans2Seg (Xie et al., 2021b), and GSD (Lin et al., 2021) and our method are in-the-wild glass segmentation methods but only rely on RGB input. From Figure 10, we can see that our method outperforms all other methods. It should be noted that our method even outperforms previous works that utilize additional input signals such as polarization cues (Xiang et al., 2021; Kalra et al., 2020; Mei et al., 2022) while being efficient.

**GSD-S dataset.** We compare our method with other recent methods in Table 15 and Figure 11, includes generic semantic segmentation methods (PSPNet (Zhao et al., 2017), DeepLabV3+ (Chen et al., 2018), PSANet (Zhao et al., 2018b), DANet (Fu et al., 2019)), recent state-of-the-art models that utilize transformer technique (SETR (Zheng et al., 2021), Swin (Liu et al., 2021b), SegFormer (Xie et al., 2021a), Twins (Chu et al., 2021)), and glass surface detection methods (GDNet (Mei et al., 2020), GSD (Lin et al., 2021), GlassSemNet (Lin et al., 2022)). For a fair comparison, all methods are retrained on the GSD-S dataset (Lin et al., 2022). Our method outperforms all other methods and achieves comparative performance with GlassSemNet (Lin et al., 2022), which has additional semantic context information. GlassSemNet (Lin et al., 2022) points out that humans frequently use the semantic context of their surroundings to reason, as this provides information about the types of things to be found and how close they might be to one another. For instance, glass windows are more likely to be found close to other semantically related objects (walls and curtains) than to things (cars and trees). So, their method utilizes semantic context information as additional input to progressively learn the contextual correlations among objects spatially and semantically, boosting their performance. Then, their predictions are refined by Fully Connected Conditional Random Fields (CRF) (Krähenbühl & Koltun, 2011) to improve their performance further.

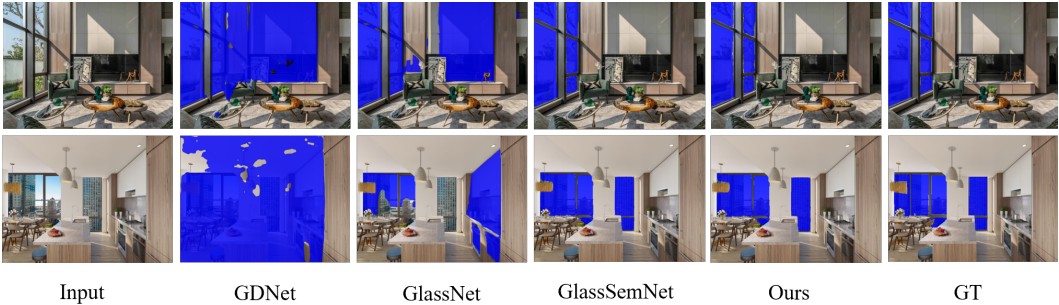

| Input | GDNet | GlassNet | GlassSemNet | Ours | GT |

Figure 11: Qualitative comparison of our method with other methods on GSD-S dataset (Lin et al., 2022).

| Method | mIoU ↑ | $F_\beta^w$ ↑ | MAE ↓ | BER ↓ |
|---|---|---|---|---|
| PSPNet (Zhao et al., 2017) | 56.1 | 0.679 | 0.093 | 13.41 |
| DeepLabV3+ (Chen et al., 2018) | 55.7 | 0.671 | 0.100 | 13.11 |
| PSANet (Zhao et al., 2018b) | 55.1 | 0.656 | 0.104 | 12.61 |
| DANet (Fu et al., 2019) | 54.3 | 0.673 | 0.098 | 14.78 |
| SCA-SOD (Siris et al., 2021) | 55.8 | 0.689 | 0.087 | 15.03 |
| SETR (Zheng et al., 2021) | 56.7 | 0.679 | 0.086 | 13.25 |
| Segmenter (Strudel et al., 2021) | 53.6 | 0.645 | 0.101 | 14.02 |
| Swin (Liu et al., 2021b) | 59.6 | 0.702 | 0.082 | 11.34 |
| Tokens-to-Token ViT (Yuan et al., 2021a) | 56.2 | 0.693 | 0.087 | 14.72 |
| SegFormer (Xie et al., 2021a) | 54.7 | 0.683 | 0.094 | 15.15 |
| Twins (Chu et al., 2021) | 59.1 | 0.703 | 0.084 | 12.43 |
| GDNet (Mei et al., 2020) | 52.9 | 0.642 | 0.101 | 18.17 |
| GSD (Lin et al., 2021) | 72.1 | 0.821 | 0.061 | 10.02 |
| Ours-B5 | 75.2 | 0.859 | 0.046 | **9.04** |
| GlassSemNet (Lin et al., 2022) † | **75.3** | **0.860** | **0.035** | 9.26 |

Table 15: Evaluation results on GSD-S dataset (Lin et al., 2022). **Note that:** we only use RGB as input for our method. The best result is **bold**, and the second best result is underline. (†) denotes the glass segmentation method with additional semantic context information and post-processing refinement.

### D.3 COMPARISON ON BINARY MIRROR SEGMENTATION

**MSD and PMD datasets.** We compare quantitative results of the state-of-the-art methods and our method on MSD and PMD datasets, including four RGB salient object detection methods CPDNet (Wu et al., 2019), MINet (Pang et al., 2020b), LDF (Wei et al., 2020), and VST (Liu et al., 2021a), and five mirror detection methods MirrorNet (Yang et al., 2019), PMDNet (Lin et al., 2020), SANet (Guan et al., 2022), VCNet (Tan et al., 2023), SATNet (Huang et al., 2023). As shown in Table 16, our method achieves the best performance in terms of all the evaluation metrics. Significantly, we outperform the second-best method by 5.63% on the MSD dataset.

**RGBD-Mirror dataset** Our method is also compared with seven RGB-D salient object detection methods such as HDFNet (Pang et al., 2020a), S2MA (Liu et al., 2020), JL-DCF (Fu et al., 2020), DANet (Fu et al., 2019), BBSNet (Fan et al., 2020) and VST (Liu et al., 2021a), and four mirror detection methods, including PDNet (Mei et al., 2021), SANet (Guan et al., 2022), VCNet (Tan et al., 2023), and PDNet (Mei et al., 2021) on the RGBD-Mirror dataset. Our method outperforms all the competing methods, even though we do not use depth information, which is shown in Table 17.

We show the visualization of all three mirror datasets in Figure 12.

| Method | MSD | | | PMD | | |
|---|---|---|---|---|---|---|
| | IoU ↑ | $F_\beta$ ↑ | MAE ↓ | IoU ↑ | $F_\beta$ ↑ | MAE ↓ |
| CPDNet (Wu et al., 2019) | 57.58 | 0.743 | 0.115 | 60.04 | 0.733 | 0.041 |
| MINet (Pang et al., 2020b) | 66.39 | 0.823 | 0.087 | 60.83 | 0.798 | 0.037 |
| LDF (Wei et al., 2020) | 72.88 | 0.843 | 0.068 | 63.31 | 0.796 | 0.037 |
| VST (Liu et al., 2021a) | 79.09 | 0.867 | 0.052 | 59.06 | 0.769 | 0.035 |
| MirrorNet (Yang et al., 2019) | 78.88 | 0.856 | 0.066 | 58.51 | 0.741 | 0.043 |
| PMDNet (Lin et al., 2020) | 81.54 | 0.892 | 0.047 | 66.05 | 0.792 | 0.032 |
| SANet (Guan et al., 2022) | 79.85 | 0.879 | 0.054 | 66.84 | 0.837 | 0.032 |
| VCNet (Tan et al., 2023) | 80.08 | 0.898 | 0.044 | 64.02 | 0.815 | 0.028 |
| SATNet (Huang et al., 2023) | 85.41 | 0.922 | 0.033 | 69.38 | 0.847 | 0.025 |
| Ours-B3 | **91.04** | **0.953** | **0.028** | **69.61** | **0.853** | **0.021** |

Table 16: Quantitative results of the state-of-the-art methods on MSD and PMD datasets. Our method achieves the best performance in terms of all the evaluation metrics.

| Method | Input | IoU ↑ | $F_\beta$ ↑ | MAE ↓ |
|---|---|---|---|---|
| HDFNet (Pang et al., 2020a) | RGB-D | 44.73 | 0.733 | 0.093 |
| S2MA (Liu et al., 2020) | RGB-D | 60.87 | 0.781 | 0.070 |
| DANet (Fu et al., 2019) | RGB-D | 67.81 | 0.835 | 0.060 |
| JL-DCF (Fu et al., 2020) | RGB-D | 69.65 | 0.844 | 0.056 |
| VST (Liu et al., 2021a) | RGB-D | 70.20 | 0.851 | 0.052 |
| BBSNet (Fan et al., 2020) | RGB-D | 74.33 | 0.868 | 0.046 |
| PDNet (Mei et al., 2021) | RGB-D | 77.77 | 0.878 | 0.041 |
| VCNet (Tan et al., 2023) | RGB | 73.01 | 0.849 | 0.052 |
| PDNet (Mei et al., 2021) | RGB | 73.57 | 0.851 | 0.053 |
| SANet (Guan et al., 2022) | RGB | 74.99 | 0.873 | 0.048 |
| SATNet (Huang et al., 2023) | RGB | 78.42 | 0.906 | 0.031 |
| Ours-B3 | RGB | **88.52** | **0.954** | **0.027** |

Table 17: Quantitative results of the state-of-the-art methods on RGBD-Mirror dataset.

## D.4 COMPARISON ON GENERIC SEGMENTATION

**Stanford2D3D datasets.** As shown in Table 18, we show more comparison with other methods in different sizes of backbones. Our method outperforms other existing works by about 10.1% better performance in mIOU. We also visualize several results of our methods in Figure 13 to highlight the segmentation capacity of our network on the general scene where there is a presence of glass objects.

**TROSD datasets.** We compared our method to other advanced techniques using the TROSD dataset (Sun et al., 2023) - a specific dataset for transparent and reflective objects. Table 19 provides an overview of our competitors and highlights their best results, achieved using their publicly available source codes. All methods utilized the same data augmentation strategy. The visualization is shown in Figure 14.

**ADE20k and Cityscapes datasets.** We conducted additional experiments on ADE20K and CityScapes datasets, with the results (mIoU) shown in Table 20 and sorted by ascending order of GFLOPs ($512 \times 512$). As can be seen, our method performs well on both datasets, with mIoU 47.5% on ADE20K and 81.9% on CityScapes.

In addition, it is feasible to integrate our model with RGB-D modalities. One approach is to incorporate an additional encoder to extract Depth features. These Depth features are combined with RGB features before being passed to the FPM module. This methodology aligns with the PDNet architecture (Mei et al., 2021) described in Table 17 and the TROSNet architecture (Sun et al., 2023) outlined in Table 19 . It is worth noting that including depth information in these approaches resulted

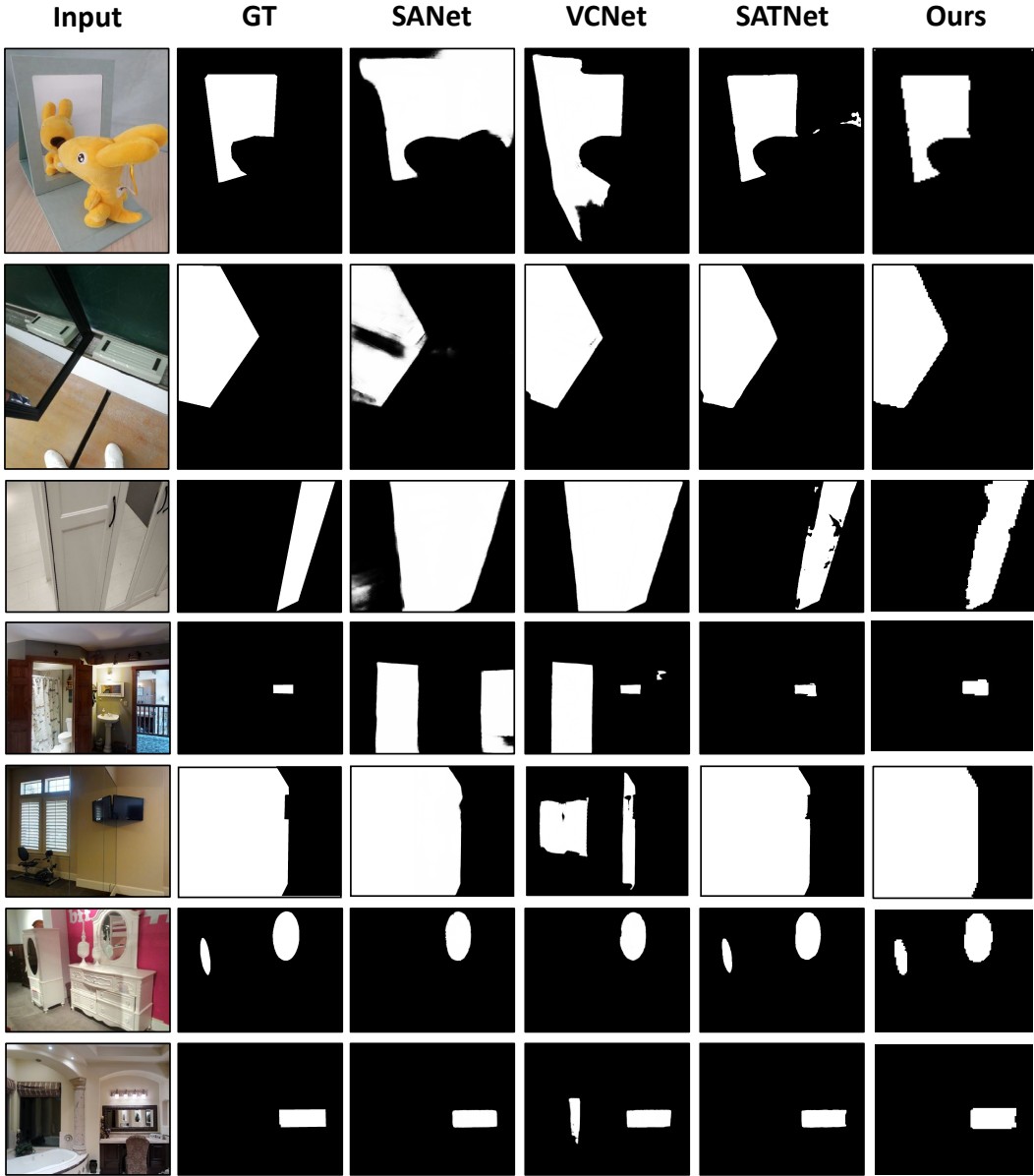

Figure 12: Qualitative comparison of our method with other methods on MSD, PMD, and RGBD-Mirror datasets.

in a notable performance improvement. Specifically, PDNet exhibited a 4% increase, while TROSNet demonstrated an 8% increase.

Finally, to fully evaluate our method's performance, we also compared our method with recent powerful foundation models such as SAM model, and the results are shown in Figure 15. It is important to note that the SAM model **does not include semantics, or in other words, it cannot yield masks with semantics**. The SAM model also presents the challenge of over-segmentation, thereby leading to a higher likelihood of false positives. As a result, we can see that the SAM model (binary and everything) can not distinguish between glass and non-glass regions compared to our method. We also tried a SAM variant with semantics (Li et al., 2023a), but their method still fails and cannot generate reasonable semantics either.

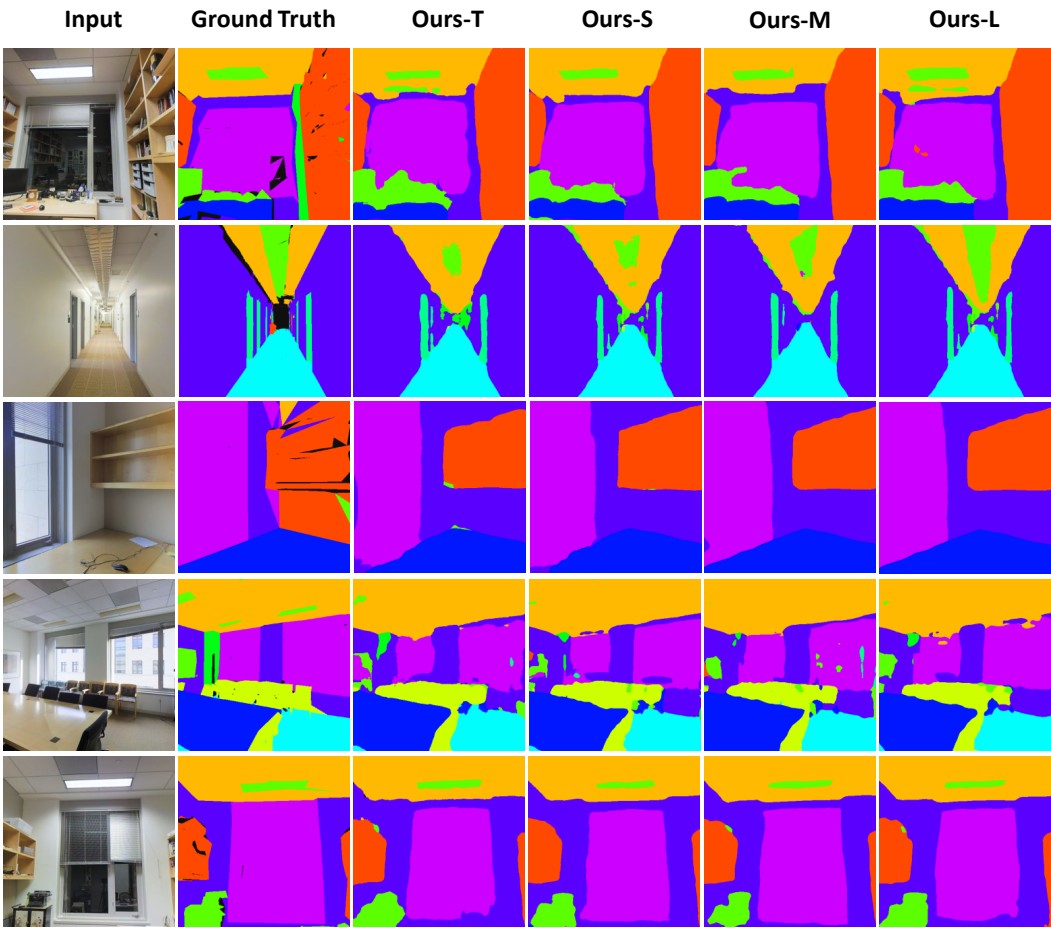

Figure 13: Visualization of our method on general scenes of Stanford2D3D dataset.

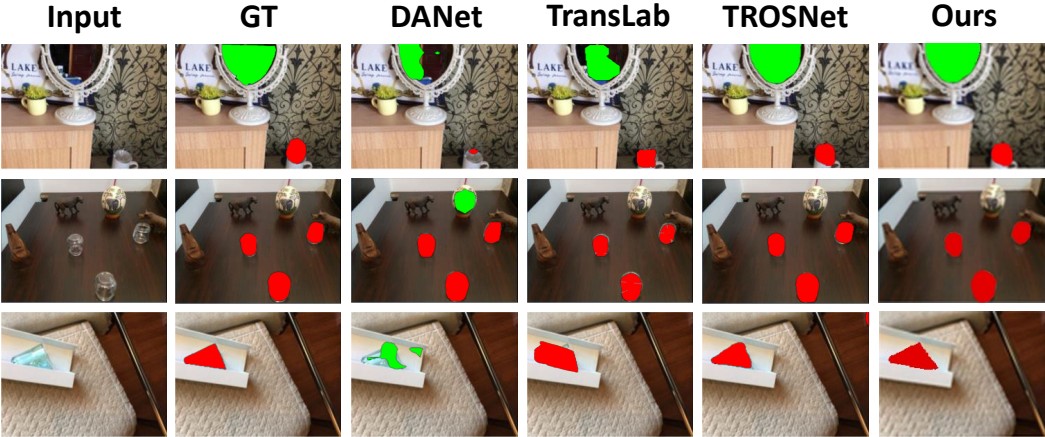

Figure 14: Qualitative comparison of our method with other methods on the TROSD dataset.

| Method | GFLOPs ↓ | MParams ↓ | mIoU ↑ |
|---|---|---|---|
| PVT-T (Wang et al., 2021b) | 10.16 | 13.11 | 41.00 |
| Trans4Trans-T (Zhang et al., 2022) | 10.45 | 12.71 | 41.28 |
| Ours-T | 10.50 | 12.72 | 47.11 |
| Trans2Seg-T (Xie et al., 2021b) | 16.96 | 17.87 | 42.07 |
| Ours-B1 | 21.99 | 14.87 | **51.55** |
| PVT-S (Wang et al., 2021b) | 19.58 | 24.36 | 41.89 |
| Trans4Trans-S (Zhang et al., 2022) | 19.92 | 23.95 | 44.47 |
| Ours-S | 20.00 | 23.98 | 50.17 |
| Trans2Seg-S (Xie et al., 2021b) | 30.26 | 27.98 | 42.91 |
| Ours-B2 | 37.03 | 27.59 | **53.98** |
| Trans4Trans-M (Zhang et al., 2022) | 34.38 | 43.65 | 45.73 |
| Ours-M | 34.51 | 43.70 | 52.57 |
| Trans2Seg-M (Xie et al., 2021b) | 40.98 | 30.53 | 43.83 |
| PVT-M (Wang et al., 2021b) | 49.00 | 56.20 | 42.49 |
| Ours-B3 | 68.35 | 51.21 | **54.66** |
| Ours-L | 50.54 | 60.86 | 53.75 |
| Ours-B4 | 79.34 | 67.11 | **55.21** |
| Ours-B5 | 154.37 | 106.19 | **55.83** |

Table 18: Comparison with state-of-the-art methods on Stanford2D3D dataset. #MParams, #GFLOPs are calculated with the input size of $512 \times 512$. The last two rows (Ours-B4, Ours-B5) show that our method can adapt to bigger backbones. **Note that:** the best result is **bold**, the second best result is underline and the results are sorting by ascending of GFLOPS.

| Method | Input | Backbone | IOU ↑ | | | mIoU ↑ | mAcc ↑ |
|---|---|---|---|---|---|---|---|
| | | | R | T | B | | |
| RefineNet (Lin et al., 2017) | RGB | ResNet-101 | 21.32 | 37.32 | 92.37 | 50.34 | 63.59 |
| ANNNet (Zhu et al., 2019) | RGB | ResNet-101 | 22.31 | 41.3 | 93.43 | 52.35 | 62.49 |
| Trans4Trans (Zhang et al., 2022) | RGB | PVTv1 | 27.69 | 39.22 | 94.16 | 53.69 | 61.82 |
| PSPNet (Zhao et al., 2017) | RGB | ResNet-101 | 26.35 | 44.38 | 94.19 | 54.97 | 64.14 |
| OCNet (Yuan et al., 2021b) | RGB | ResNet-101 | 31.76 | 46.52 | 95.05 | 57.78 | 64.46 |
| TransLab (Xie et al., 2020) | RGB | ResNet-50 | 42.57 | 50.72 | 96.01 | 63.11 | 68.72 |
| DANet (Fu et al., 2019) | RGB | ResNet-101 | 42.76 | 54.39 | 95.88 | 64.34 | 70.95 |
| TROSNet (Sun et al., 2023) | RGB | ResNet-50 | 48.75 | 48.56 | 95.49 | 64.26 | 75.93 |
| Ours | RGB | PVTv2 | **66.16** | **66.83** | **97.71** | **76.90** | **87.62** |
| SSMA (Valada et al., 2019) | RGB-D | ResNet-50 | 24.7 | 29.04 | 89.98 | 47.91 | 67.72 |
| FRNet (Zhou et al., 2022) | RGB-D | ResNet-34 | 28.37 | 36.59 | 92.18 | 52.38 | 63.94 |
| EMSANet (Seichter et al., 2022) | RGB-D | ResNet-101 | 27.53 | 44.1 | 96.14 | 55.92 | 71.63 |
| FuseNet (Hazirbas et al., 2016) | RGB-D | VGG-16 | 37.3 | 43.29 | 94.97 | 58.52 | 66.13 |
| RedNet (Jiang et al., 2018) | RGB-D | ResNet-50 | 48.27 | 47.57 | 95.76 | 63.87 | 69.23 |
| EBLNet (He et al., 2021) | RGB-D | ResNet | 51.75 | 50.12 | 94.57 | 65.49 | 67.39 |
| TROSNet (Sun et al., 2023) | RGB-D | ResNet-50 | 62.27 | 57.23 | 96.52 | 72.01 | 81.21 |

Table 19: Performance comparison of different methods on TROSD. R: reflective objects. T: transparent objects. B: background. The best results are in bold.

| Method | GFLOPs↓ | MParams↓ | ADE20K | CityScapes |
|---|---|---|---|---|
| Trans4Trans-M (PVTv2-B3) (Wang et al., 2022) | 41.9 | 49.6 | - | 69.3 |
| Semantic FPN (PVTv2-B3) (Zhang et al., 2022) | 62.4 | 49.0 | 47.3 | - |
| Ours-B3 (PVTv2-B3) | 68.3 | 51.2 | 47.5 | 81.9 |
| MogaNet-S (SemFPN) (Li et al., 2023b) | 189 | 29.0 | 47.7 | - |
| NAT-Mini (UPerNet) (Hassani et al., 2023) | 900 | 50.0 | 46.4 | - |
| InternImage-T (UPerNet) (Wang et al., 2023) | 944 | 59.0 | 47.9 | 82.5 |

Table 20: Quantitative results of the state-of-the-art methods on ADE20k and Cityscapes datasets.

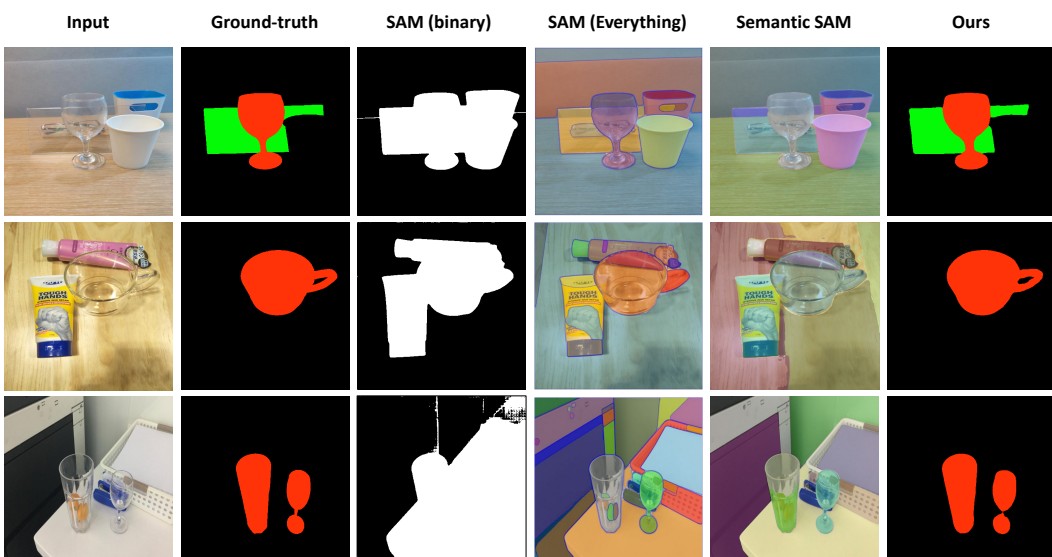

Figure 15: Qualitative comparison of our method with recent foundation models on Trans10k-v2 dataset.

## D.5 FURTHER ANALYSIS

To provide a theoretical verification of the effectiveness of the RFE module, we conducted the following additional experiments:

- We take a model with the RFE module that has already been trained. To prove that RFE is effective, we compare the feature maps before and after passing through the RFE module. Please see Figure 16 below. In our example, we can see that after passing through the RFE module, we can get a stronger reflection signal, such as the transparent glass area or the specular reflection appearing at the base of the wine glass.
- Using the same model, we try disabling the RFE module at inference by passing the feature map before RFE directly to the next step. Note that at training, the RFE module is well-trained as usual. Figure 17 shows that bypassing RFE results in a noisy feature map and wrong mask prediction. This means that our learning of RFE does not yield a trivial function, e.g., identity, and RFE does play an important role in processing the feature maps and output reflection masks.

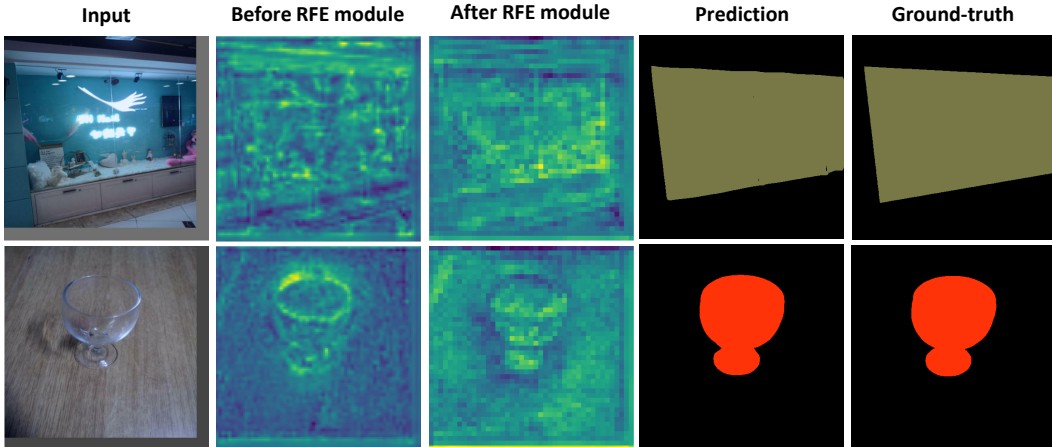

Figure 16: Comparison of the feature maps before and after passing through the RFE module on Trans10k-v2 dataset.

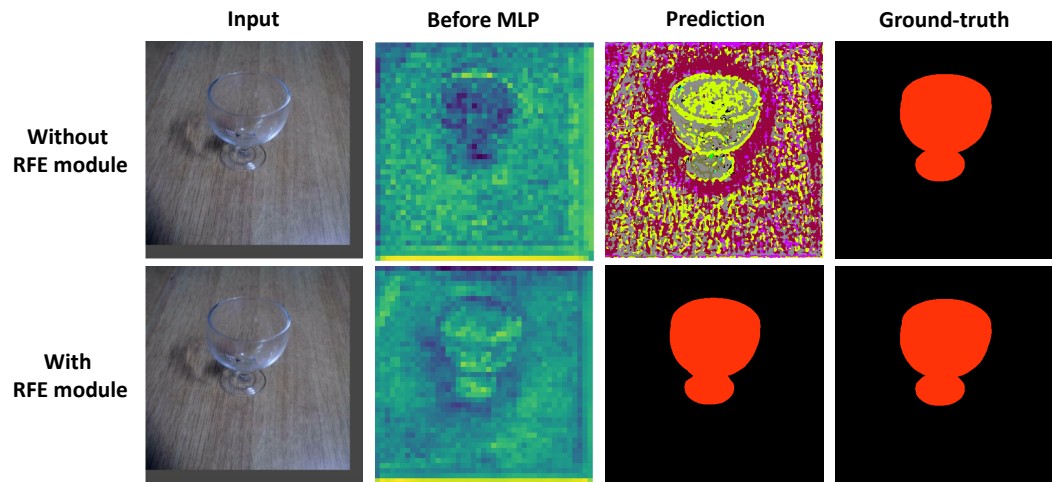

Figure 17: Comparison of the feature maps at inference by passing through the RFE module as usual (top row) and bypassing the RFE module (bottom row) on Trans10k-v2 dataset. Note that at training, the RFE module is well-trained as usual.

