# OpenReview forum: "TransCues: Boundary and Reflection-empowered Pyramid Vision Transformer for Semantic Transparent Object Segmentation"
_ICLR.cc/2024/Conference — Submitted to ICLR 2024_

### Official Review · Reviewer_42YL · 2023-10-15

**Soundness:** 3 good
**Presentation:** 3 good
**Contribution:** 2 fair
**Rating:** 5
**Confidence:** 5

**Summary:**

This paper presents a hierarchical architecture for transparent object segmentation. Boundary and reflection cues are incorporated in the module designs. Extensive experiments are conducted on multiple benchmarks, which shows the effectiveness of the proposed model. The paper is overall well-written and nicely structured.

**Strengths:**

1. The proposed model achieves state-of-the-art performances on multiple datasets.
2. Failure cases are well studied.
3. The paper is overall well-written and nicely structured.

**Weaknesses:**

1. In Fig. 5, does it show that the proposed method is consistently effective for different backbones? This should be better discussed.
2. In Table 6, it would be nice to show the computation complexity of the two designed modules for analysis.
3. How to theoretically verify that the proposed method did really make use of reflection cues? This could be better discussed.
4. It is hard to find any novel operations in the proposed reflection feature enhancement module as it simply combines existing mechanisms. It would be nice to clarify the technical novelty and theoretical contributions of the proposed modules.
5. There are extensive segmentation methods that introduce boundary-relevant loss designs or other designs. Please consider incorporating some existing boundary-relevant designs for a comparison. This can better show the superiority of your proposed boundary feature enhancement module.
6. The related work follows that of Trans4Trans. It would be nice to add more recent related state-of-the-art works.

Sincerely,

**Questions:**

Would it be possible to incorporate your model with the RGB-D modalities for an experiment? This could be discussed.

When the proposed model works on images without any transparent objects, would it create false positives? This could be assessed.

Sincerely,

---

> ### Author Response · Authors · 2023-11-20
> **To Reviewer 42YL (1/2)**
>
> **Q1** In Fig. 5, does it show that the proposed method is consistently effective for different backbones? This should be better discussed.
>
>  > **A1** The relationship between the complexity, size, and performance of the backbone is shown in Figure 5. It is evident that there is a clear trade-off between the three factors. The performance improves as the backbone size increases. This relationship is crucial for users as they need to balance the computational demands with the performance requirements of their specific applications. Our research findings suggest that choosing a larger backbone leads to better performance, which provides a strategic advantage for those aiming to enhance the effectiveness of their models.
>
>  **Q2** In Table 6, it would be nice to show the computation complexity of the two designed modules for analysis.
>
>  > **A2** The below is the updated version for Table 6. We added to the revised paper.
>  > | **Backbone** | **GFLOPs&#8595;** | **MParams&#8595;** | **BFE** | **RFE** | **Stanford2D3D (mIoU) &#8593;** | **Trans10K-v2 (mIoU) &#8593;**  |
>  > |--------------|:------------:|:-------------:|:------------:|:------------:|------------------------------------------------------|------------------------------------------------------|
>  > | PVTv1-T      | 10.16 | 13.11 | -          | -          | 45.19                                                | 69.44                                                |
>  > | PVTv2-B1     | 11.48 | 13.89 | -          | -          | 46.79 +1.6          | 70.49 +1.05          |
>  > | PVTv2-B1     | 13.22 | 14.37 | -          | &check; | 48.12 +2.93          | 72.65 +3.21          |
>  > | PVTv2-B1     | 19.55 | 14.39 | &check; | -          | 50.22 +5.03         | 74.89 +5.45 |
>  > | PVTv2-B1     | 21.29 | 14.87 | &check; | &check; | 51.55 +6.36| 77.05 +7.61 |
>
>
>  **Q3** How to theoretically verify that the proposed method did really make use of reflection cues? This could be better discussed.
>
>  > **A3** As depicted in Figure 2 and quantified in Table 6, our method effectively extracts regions of reflection, substantially improving overall performance. A potential enhancement to this approach could involve incorporating a binary classifier responsible for identifying the lack of reflections before activating the RFE module. This preliminary classification step has the potential to conserve computational resources by determining when no reflections are present and deciding to bypass the RFE process, thus optimizing the method's efficiency.
>
>
>  **Q4** It is hard to find any novel operations in the proposed reflection feature enhancement module as it simply combines existing mechanisms. It would be nice to clarify the technical novelty and theoretical contributions of the proposed modules.
>
>  > **A4** The RFE module utilizes an extensive convolution-deconvolution architecture to process input data and generate an improved feature map. The system takes in input features and carefully processes them to generate a sophisticated feature map. The utilization of a layered architecture is of utmost importance, as it effectively handles visual data at various levels of abstraction, effectively navigating the complexities associated with diverse visual occurrences, including reflections. In contrast to models designed for global reflection scenarios, which assume that reflections are present throughout the entire image, our RFE module demonstrates superior performance in detecting local reflections. Implementing a localized technique is of utmost importance in real-world scenarios characterized by irregular and area-specific reflections rather than uniform ones. The RFE module demonstrates high accuracy in detecting glass surfaces within various environmental conditions, primarily by prioritizing the analysis of localized reflections. Hence, this approach offers a more accurate and contextually nuanced method for discerning reflection within the multifaceted fabric of real-world environments.

---

> > ### Author Response · Authors · 2023-11-20
> > **To Reviewer 42YL (2/2)**
> >
> > **Q5** There are extensive segmentation methods that introduce boundary-relevant loss designs or other designs. Please consider incorporating some existing boundary-relevant designs for a comparison. This can better show the superiority of your proposed boundary feature enhancement module.
> >
> >  > **A5** Let's consider some recent methods that incorporate boundary-relevant designs, such as **[Tian et al., 2022]** and **[Lin et al., 2020]**. These papers incorporate the Canny filter into their methodologies, albeit for different purposes. **[Tian et al., 2022]** employs the Canny operator to enhance image boundary delineation. In contrast, **[Lin et al., 2020]** use it to extract precise mirror edges from ground truth masks to create supervisory edge maps. The Canny filter is celebrated for its ability to generate edges that are not only thin and smooth but also notably accurate. This precision stems from using Non-maxima suppression and finely adjustable parameters to cater to varying image characteristics. However, these advantages come at the cost of increased complexity and computational demand compared to simpler alternatives like the Sobel filter. Furthermore, the Canny filter's reliance on manual parameter adjustment necessitates extensive experimentation to ascertain optimal settings, as thoroughly examined in a dedicated section of **[Tian et al., 2022]**. In contrast, our decision to use the Sobel filter is informed by our commitment to efficiency, particularly in applications that require real-time processing, such as robotic navigation. With its relative simplicity and lower computational load, the Sobel filter aligns with our objectives to develop a framework that balances performance with practicality. This choice underscores our framework's suitability for scenarios where speed and efficiency are paramount without significantly compromising the edge detection quality needed for effective navigation.
> >  >
> >  > **[Tian et al., 2022]** Xin Tian, Ke Xu, Xin Yang, Baocai Yin and Rynson W. H. Lau. Learning to Detect Instance-Level Salient Objects Using Complementary Image Labels. In IJCV, 2022.
> >  >
> >  > **[Lin et al., 2020]** Lin Jiaying, Wang Guodong and Lau Rynson W.H. Progressive Mirror Detection. In CVPR, 2020.
> >
> >  **Q6** The related work follows that of Trans4Trans. It would be nice to add more recent related state-of-the-art works.
> >
> >  > **A6** We have revised Section 2 as suggested, please check the revised version (with highlighted changed text)
> >
> >  **Q7** Would it be possible to incorporate your model with the RGB-D modalities for an experiment? This could be discussed.
> >
> >  > **A7** Indeed, it is feasible to integrate our model with RGB-D modalities. One approach is to incorporate an additional encoder to extract Depth features. These Depth features are combined with RGB features before being passed to the FPM module. This methodology aligns with the PDNet architecture **[Mei et al., 2021]** described in Table 17 and the TROSNet architecture **[Sun et al., 2023]** outlined in Table 19. It is worth noting that including depth information in these approaches resulted in a notable performance improvement. Specifically, PDNet exhibited a $4\%$ increase, while TROSNet demonstrated an $8\%$ increase. We also added this discussion to the Section D.4 of the revised paper.
> >  >
> >  > **[Mei et al., 2021]** Haiyang Mei, Bo Dong, Wen Dong, Pieter Peers, Xin Yang, Qiang Zhang, and Xiaopeng Wei. Depth-aware mirror segmentation. In CVPR, 2021.
> >  >
> >  > **[Sun et al., 2023]** Tianyu Sun, Guodong Zhang, Wenming Yang, Jing-Hao Xue, and Guijin Wang. Trosd: A new rgb-d dataset for transparent and reflective object segmentation in practice. IEEE TCSVT, 2023.
> >
> >  **Q8** When the proposed model works on images without any transparent objects, would it create false positives? This could be assessed.
> >
> >  > **A8** Our approach is designed to effectively handle the challenges of detecting transparent and reflective items in real scenarios. We evaluated our method against a wide range of tasks, FOVs, and object placements, as outlined in Section C.1, and tested it on the diverse Stanford2D3D dataset, which includes a small proportion of transparent objects (around $1\%$). As detailed in Section 4.1, our method outperforms current methods, achieving an **$8.25\%$** increase in mIOU for semantic scene segmentation, demonstrating its enhanced ability to distinguish between different objects and settings. We conducted additional experiments on two standard semantic segmentation datasets: ADE20K and CityScapes (which have less than $10\%$ of transparent objects). Our method performs well on both datasets, with mIoU **$47.5\%$** on ADE20K and **$81.9\%$** on CityScapes, as shown in Table 19 and Section D.4.

---

> > > ### Comment · Reviewer_42YL · 2023-11-22
> > > **Comment on A5 and A3**
> > >
> > > Regarding the comparison mentioned in A5, if it is possible, please consider adding some numerical results of their performances. The responses in A3 do not fully present theoretical verifications of the exploitation of reflection cues for transparency perception.
> > >
> > > The reviewer would like to thank the authors for their responses and added analyses.
> > >
> > > Sincerely,

---

> > > > ### Author Response · Authors · 2023-11-23
> > > > **Response to the Comment on A3 and A5**
> > > >
> > > > Thank you for getting back to us. We appreciate your response. Below, we will be addressing the concerns you have raised.
> > > >
> > > > **Q3** The responses in A3 do not fully present theoretical verifications of the exploitation of reflection cues for transparency perception.
> > > > > To provide a theoretical verification of the effectiveness of the RFE module, we conducted the following additional experiments:
> > > > > * We take a model with the RFE module that has already been trained. To prove that RFE is effective, we compare the feature maps before and after passing through the RFE module. Please see **Figure 16** of the revised paper. In our example, we can see that after passing through the RFE module, we can get a stronger reflection signal, such as the transparent glass area or the specular reflection appearing at the base of the wine glass.
> > > > >
> > > > > * Using the same model, we try disabling the RFE module at inference by passing the feature map before RFE directly to the next step. Note that at training, the RFE module is well-trained as usual. **Figure 17** of the revised paper shows that bypassing RFE results in a noisy feature map and wrong mask prediction. This means that our learning of RFE does not yield a trivial function, e.g., identity, and RFE does play an important role in processing the feature maps and output reflection masks.
> > > >
> > > > **Q5** Regarding the comparison mentioned in A5, if it is possible, please consider adding some numerical results of their performances.
> > > > > * Currently, we cited [Tian et al., 2022] following the suggestion of Reviewer 3ba2. Unfortunately, their source code is not publicly available, and their experiments did not cover the same datasets as ours due to their emphasis on a distinct salient instance detection task. Hence, a direct comparison is not feasible.
> > > > >
> > > > > * In Table 16, a comparison was made between the PMD method [Lin et al., 2020] and our approach, revealing a significant performance advantage on our part, such as **81.54%** vs **91.04%** mIoU and **66.05%** vs **69.61%** mIoU on MSD and PMD datasets, respectively.

---

### Official Review · Reviewer_3ba2 · 2023-10-17

**Soundness:** 3 good
**Presentation:** 3 good
**Contribution:** 3 good
**Rating:** 6
**Confidence:** 5

**Summary:**

This paper proposes the TransCues, a transformer encoder-decoder network for the segmentation of glass, mirrors, and transparent objects. The main idea of this paper is to model the boundary and the reflection cues. Accordingly, a Boundary Feature Enhancement (BFE) module and a Reflection Feature Enhancement (RFE) module are proposed. The BFE module is implemented based on the ASPP module and the RFE module has an encoder-decoder structure. The paper runs experiments on eight existing datasets, and the comparisons show that the proposed method achieves impressive results, but with different models.

**Strengths:**

The paper has certain merits.
Although the boundary and reflection cues have been explored in previous works, the paper shows that a better network design that focuses on low-level features may improve the segmentation of mirrors and glass surfaces/objects.
The paper provides extensive comparisons on eight benchmarks, which shows an overall picture of this topic.
The paper is generally easy to read and understand.

**Weaknesses:**

However, I have some concerns.
The first concern is about the results. The paper creates a lot of models, I.e., TransCues -T, -S, -M, -L, -B1, -B2, -B3, -B4, -B5, while some of them are based on PVTv1, and the others are based on PVTv2. During the comparisons, Table 1 uses B4, Table 2 and 5 use B2, Table 3 and 4 use B3, and the Table 6 uses B1. This makes the comparisons very messy, which may not provide meaningful analysis/discussions. What are the criterion of such selections? I note that there are only one Table (Table 13 in the supplemental) includes all nine TransCues models, from which it seems that B1 and B2 outperforms Ours-L with less parameters. How often and why does this happen is not known.
The Abstract mentions that the RFE module ``decomposes reflections into foreground and background layers’’, however, in section 3.3, I do not find corresponding designs and the motivations of such designs. Second, section 3.4 uses pseudo ground truth reflection masks, but it is not mentioned how these pseudo labels are created. Third, the paper only discuss RFE with (Zhang et al., 2018) regarding the reflection modeling. The ICCV’21 paper ``Location-aware Single Image Reflection Removal’’ detects the strong reflections. Would it be better to use reflection removal methods to generate pseudo labels?
The boundary loss seems not a novelty. If so, I suggest to move it onto the supplemental. Otherwise, the paper needs to explain where the novelty is and provides discussions with existing methods. For example, the IJCV’22 paper ``Learning to Detect Instance-level Salient Objects Using Complementary Image Labels’’ uses canny operators to enhance the boundary information. The PMD (Lin et al. 2020) also uses ground truth boundary information for the supervision.
The feature flow in the RFE module (Figure 7 of supp.) is rather complicated and more explanation is helpful, in order to evaluate its novelty.
The placements of RFE and BFE seems casual. I can only guess the reason might be that the authors try to focus the whole network on low-level features. More explanation is helpful.
The ablation study only includes the RFE and BFE, while it is not known how much contributions the FEM, FPM and the final MLP have made to the segmentation performance.
The model relies on the detection of reflections, while for glass surface/objects segmentation, the question is whether reflections can always be detected, and if not, how does it affects the final results? The paper shows failure cases on the Trans10K-v2, but such cases seem dataset-specific. It is better to show failure cases that caused by the limitations of the proposed model.




Below are some suggestions.
Use the symbols (including the use of, e.g., \mathcal) more consistently.
The Position Embedding and the Encoder and Decoder paragraphs in section 3.1 can be shortened.

%%%%%%%%%%%%%%%%%%%%%%%%%%%%%%%%%%%%%%%%%%%%%%%%%

I have read the responses. My previous concerns are answered but some of them are not addressed well. One suggestion is to include all models (TransCues -T, -S, -M, -L, -B1, -B2, -B3, -B4, -B5) in one table somewhere at the beginning (with their numbers of parameters listed) so that readers can see an overall picture. The other suggestion is to provide some experimental results/statistics regarding Q5 and Q7 in their revision or re-submission.

**Questions:**

Please see above.

---

> ### Author Response · Authors · 2023-11-20
> **To Reviewer 3ba2 (1/3)**
>
> **Q1** The first concern is about the results. The paper creates a lot of models, i.e., TransCues -T, -S, -M, -L, -B1, -B2, -B3, -B4, -B5, while some of them are based on PVTv1, and the others are based on PVTv2. During the comparisons, Table 1 uses B4, Table 2 and 5 use B2, Table 3 and 4 use B3, and the Table 6 uses B1. This makes the comparisons very messy, which may not provide meaningful analysis/discussions. What are the criterion of such selections? I note that there are only one Table (Table 13 in the supplemental) includes all nine TransCues models, from which it seems that B1 and B2 outperforms Ours-L with less parameters. How often and why does this happen is not known.
>
>  > **A1** The key criterion for our model selection is to ensure fair comparisons. For each experiment, we have carefully selected a our model variant that has similar model size used by other methods, as indicated in the respective tables. Due to the rich adaptation of transformers in practice, previous methods have used several different backbones with different sizes. It would be unfair to compare a large model to a small model as their performance and number of parameters are very different. For deployment in a real use case, one should choose a model that fits their computational budget on their platform.
>
>
>
>  **Q2** The Abstract mentions that the RFE module decomposes reflections into foreground and background layers, however, in section 3.3, I do not find corresponding designs and the motivations of such designs. Second, section 3.4 uses pseudo ground truth reflection masks, but it is not mentioned how these pseudo labels are created. Third, the paper only discuss RFE with (Zhang et al., 2018) regarding the reflection modeling. The ICCV’21 paper Location-aware Single Image Reflection Removal’’ detects the strong reflections. Would it be better to use reflection removal methods to generate pseudo labels?
>
>  > **A2**
>  > * Apologies for this confusion and thank you for pointing this out. In Section 3.3, we meant that it is possible to separate the reflection area (foreground) from the non-reflection area (background). The mechanics of this process are depicted in Figure 7, where we detail the workings of the RFE module. We can see that after the last deconvolution layer, we split the features into two parts: the enhanced features (with reflection property)  and the reflection mask area (which is supervised by pseudo ground truth). This bifurcation is pivotal in our model's ability to identify and segregate reflective surfaces from their surroundings accurately. We will revise the writing to improve the clarity on this point.
>  > * Section 3.4: The $\phi()$ function function employs an erosion operation on the semantic mask to isolate potential reflection areas that are crucial for our model's accurate discernment of reflective surfaces (we assume that common categories like window, door, cup, bottle, etc will have reflective appearance).
>  > * Regrading to utilizing reflection removal methods, it can help to generate pseudo labels can offer certain advantages. However, it is crucial to acknowledge that these methods primarily focus on mitigating global reflections, which occur when the presence of glass encompasses the entirety of an image. The approach above demonstrates limitations when applied to intricate real-world situations, particularly those involving glass objects distributed across the scene rather than occupying a dominant position. The RFE module in our study presents a comparison of different approaches. It can detect localized reflections and distinguish glass surfaces based on the semantic mask. It is better suited to the diverse and unpredictable conditions found in real-world situations, where reflections are specific to certain areas rather than uniformly distributed over the entire image. As a result, the RFE module provides a more accurate and contextually appropriate approach for identifying reflections in real-world environments.

---

> > ### Author Response · Authors · 2023-11-20
> > **To Reviewer 3ba2 (2/3)**
> >
> > **Q3** The boundary loss seems not a novelty. If so, I suggest to move it onto the supplemental. Otherwise, the paper needs to explain where the novelty is and provides discussions with existing methods. For example, the IJCV’22 paper ''Learning to Detect Instance-level Salient Objects Using Complementary Image Labels'' uses canny operators to enhance the boundary information. The PMD (Lin et al. 2020) also uses ground truth boundary information for the supervision.
> >
> >  > **A3** We will address the concern of boundary loss as below:
> >  > * The Sobel image kernel, sometimes called the Sobel-Feldman filter, is widely used in image processing and computer vision. Primarily employed in edge detection algorithms, it draws attention to the boundaries of an image. To do this, we analyze the 2D gradient of an image and highlight its high spatial frequency regions. To calculate the gradient magnitude at each pixel, the Sobel kernel approximates the derivative of the image along both the x and y axes. It then combines these approximations to find the overall gradient. Specifically, it calculates the convolution of the image with two small, separable filters designed to highlight changes in intensity along each axis. So, our Boundary loss $L_{\textit{boundary}}$ based on Sobel filter to measure how closely the gradients of a predicted mask match the gradients of the GT mask using Dice loss. In image segmentation tasks, Dice loss is usually chosen to determine whether the model achieves satisfactory results, which is achieved by calculating the similarity of the predicted samples to the real samples (background truth). Also, the detected boundary is too small, thus resulting in highly imbalanced positive and negative samples. Therefore, utilizing Dice loss is a better solution than BCE loss of the two mentioned above methods.
> >  > * The IJCV and PMD papers incorporate the Canny filter into their methodologies, albeit for different purposes. The IJCV paper employs the Canny operator to enhance boundary delineation within images. In contrast, the PMD paper uses it to extract precise mirror edges from ground truth masks to create supervisory edge maps. The Canny filter is celebrated for its ability to generate edges that are not only thin and smooth but also notably accurate. This precision stems from using Non-maxima suppression and finely adjustable parameters to cater to varying image characteristics. However, these advantages come at the cost of increased complexity and computational demand compared to simpler alternatives like the Sobel filter. Furthermore, the Canny filter's reliance on manual parameter adjustment necessitates extensive experimentation to ascertain optimal settings, as thoroughly examined in a dedicated section within the IJCV paper. In contrast, our decision to use the Sobel filter is informed by our commitment to efficiency, particularly in applications that require real-time processing, such as robotic navigation. With its relative simplicity and lower computational load, the Sobel filter aligns with our objectives to develop a framework that balances performance with practicality. This choice underscores our framework's suitability for scenarios where speed and efficiency are paramount without significantly compromising the edge detection quality needed for effective navigation.
> >
> >
> >  **Q4** The feature flow in the RFE module (Figure 7 of supp.) is rather complicated and more explanation is helpful, in order to evaluate its novelty.
> >
> >  > **A4** We have revised the text and figure of the RFE module to improve clarity.
> >
> >  **Q5** The placements of RFE and BFE seems casual. I can only guess the reason might be that the authors try to focus the whole network on low-level features. More explanation is helpful.
> >
> >  > **A5** The placement of RFE and BFE is based on the fact that we prioritize low-level features, a choice substantiated by our empirical findings. This strategic focus aims to enhance performance while reducing the overall model size, where we show that a smaller model does not compromise efficacy much. Such insights result from rigorous analysis and validation, which are thoroughly documented and discussed in our study.

---

> > > ### Author Response · Authors · 2023-11-20
> > > **To Reviewer 3ba2 (3/3)**
> > >
> > > **Q6** The ablation study only includes the RFE and BFE, while it is not known how much contributions the FEM, FPM and the final MLP have made to the segmentation performance.
> > >
> > >  > **A6** We apologize for any confusion caused by our explanation in Section 4.2. In our ablation study, the baseline models are constructed with two modules (FEM and FPM). The FEM is an adaptation of the PVT-v2 architecture, specifically modified by omitting its final classification layer to suit our needs. On the other hand, the FPM is a streamlined and lighter version of the FEM, deliberately designed to be lighter and more efficient than utilizing standard decoders like Semantic FPN, SegFormer, UPerNet, and the like. These architectural modifications are meticulously illustrated in Figure 6. The final MLP layer is utilized as a learnable segmentation head to predict per-pixel labels.
> > >
> > >  **Q7** The model relies on the detection of reflections, while for glass surface/objects segmentation, the question is whether reflections can always be detected, and if not, how does it affects the final results?
> > >
> > >  > **A7** In Figure 2 and Table 6, we have shown that our method can extract well the reflection areas and utilize it to improve the overall performance. However, if the reflection on the glass surface exhibits insufficient strength (poor or weak) to be discerned by our RFE module, our model may encounter challenges in accurately detecting the glass surface. It is worth noting that in this particular scenario, the correct identification of glass surfaces poses a challenge, even for human observers.
> > >
> > >  **Q8** The paper shows failure cases on the Trans10K-v2, but such cases seem dataset-specific. It is better to show failure cases that caused by the limitations of the proposed model.
> > >
> > >  > **A8** We added more visualizations to Figure 4 as suggested.
> > >
> > >  **Q9** The Position Embedding and the Encoder and Decoder paragraphs in section 3.1 can be shorten.
> > >
> > >  > **A9** We have revised Section 3.1 as suggested, please check the revised version (with highlighted changed text).

---

### Official Review · Reviewer_WwQc · 2023-11-06

**Soundness:** 2 fair
**Presentation:** 3 good
**Contribution:** 2 fair
**Rating:** 5
**Confidence:** 4

**Summary:**

The paper introduces an efficient transformer-based segmentation architecture TransCues, which exhibits strong performance in segmenting transparent objects. This capability is attributed to the innovative integration of the Boundary Feature Enhancement module and the Reflection Feature Enhancement module.
The authors show solid results on various transparent object segmentation and generic semantic segmentation benchmarks and conducts comprehensive ablation studies on their core design choices.

**Strengths:**

The content is well-organized and easy to follow. The motivation is well-established and the effectiveness of their solution is verified by extensive experiment. The proposed architecture achieved competitive performance on a wide range of tasks, while maintaining competitive efficiency.

**Weaknesses:**

The authors regard the boundary loss as their contribution, but do not provide an ablation of this module. Similarly, the reflection loss also has not been ablated.
The authors claim that their proposed approach is robust to generic semantic segmentation tasks, but do not evaluate on the most widely used semantic segmentation datasets, such as ADE20K and cityscapes.
The influence of different pretraining of the backbone is not properly assessed;
The authors claim that most semantic segmentation models struggle to distinguish between glass and non-glass regions, but does this assertion still hold true for the state of the art generic semantic segmentation model, such as SAM?

**Questions:**

see weakness

---

> ### Author Response · Authors · 2023-11-20
> **To Reviewer WwQc (1/2)**
>
> **Q1&2** The authors regard the boundary loss as their contribution, but do not provide an ablation of this module. The reflection loss also has not been ablated.
>
>  > **A1&2** We provided the ablation studies in Section 4.2 (Effectiveness of different modules) in the paper. As shown in Figure 7, the boundary loss and the reflection loss are tied to the network architecture with BFE and RFE modules. Therefore, the ablation studies with these losses are equivalent to ablation studies on the BFE and RFE modules.
>
>  **Q3** The authors claim that their proposed approach is robust to generic semantic segmentation tasks, but do not evaluate on the most widely used semantic segmentation datasets, such as ADE20K and cityscapes.
>
>  > **A3** Selecting which dataset to perform the experiments has been an important topic of discussion when our team developed this paper. To rigorously assess the effectiveness of our proposed method, we have curated a diverse array of datasets that encompass a broad spectrum of tasks, fields of view (FOV), and object placements, as detailed in Section C.1. We have conducted extensive testing on the Stanford2D3D dataset as this dataset reflects well real-world scenarios, where common objects dominate and transparent objects are less than $1\%$ of the total dataset.
>  >
>  > As suggested by the reviewer, we conducted additional experiments on ADE20K and CityScapes datasets, with the results (**mIoU**) shown below and were added to Section D.4 and Table 19. Note that the results of other methods are taken from [Paperswithcode-ADE20k](https://paperswithcode.com/sota/semantic-segmentation-on-ade20k), [Paperswithcode-CityScapes](https://paperswithcode.com/sota/semantic-segmentation-on-cityscapes) and their manuscripts, and sorted by ascending order of GFLOPs ($512 \times 512$).
>  >
>  > As can be seen, our method performs well on both datasets, with mIoU **$47.5\%$** on ADE20K and **$81.9\%$** on CityScapes.
>  >
>  > | Method                                             | GFLOPs&#8595; | MParams&#8595; | ADE20K (**mIoU**) &#8593; |  CityScapes (**mIoU**) &#8593; |
>  > |:-------------------------------------------------- |:-------------:|:--------------:|:------:|:------------:|
>  > | Trans4Trans-M (PVTv2-B3) **[Zhang et al., 2022]**  |     41.9      |      49.6      |    -   |     69.3     |
>  > | Semantic FPN (PVTv2-B3) **[Wang et al., 2022]**    |     62.4      |      49.0      |  47.3  |      -       |
>  > | Ours-b3 (PVTv2-B3)                                 |     68.3      |      51.2      |  47.5  |     81.9     |
>  > | MogaNet-S (SemFPN)   **[Li et al., 2023]**         |     189       |      29.0      |  47.7  |      -       |
>  > | NAT-Mini (UPerNet) **[Hassani et al., 2023]**      |     900       |      50.0      |  46.4  |      -       |
>  > | InternImage-T (UPerNet) **[Wang et al., 2023]**    |     944       |      59.0      |  47.9  |     82.5     |
>  >
>  > **[Wang et al., 2022]** Wenhai Wang, Enze Xie, Xiang Li, Deng-Ping Fan, Kaitao Song, Ding Liang, Tong Lu, Ping Luo, and Ling Shao. Pvtv2: Improved baselines with pyramid vision transformer. In Computational Visual Media, 2022.
>  >
>  > **[Zhang et al., 2022]** Jiaming Zhang, Kailun Yang, Angela Constantinescu, Kunyu Peng, Karin M¨uller, and Rainer Stiefelhagen. Trans4trans: Efficient transformer for transparent object and semantic scene segmentation in real-world navigation assistance. In IEEE T-ITS, 2022.
>  >
>  > **[Li et al., 2023]** Siyuan Li, Zedong Wang, Zicheng Liu, Cheng Tan, Haitao Lin, Di Wu, Zhiyuan Chen, Jiangbin Zheng, Stan Z. Li. Efficient Multi-order Gated Aggregation Network. In arXiv 2211.03295v2, 2023.
>  >
>  > **[Hassani et al., 2023]** Ali Hassani and Steven Walton and Jiachen Li and Shen Li and Humphrey Shi. Neighborhood Attention Transformer. In CVPR, 2023.
>  >
>  > **[Wang et al., 2023]** Wang, Wenhai and Dai, Jifeng and Chen, Zhe and Huang, Zhenhang and and others. InternImage: Exploring Large-Scale Vision Foundation Models with Deformable Convolutions. In CVPR, 2023.

---

> > ### Author Response · Authors · 2023-11-20
> > **To Reviewer WwQc (2/2)**
> >
> > **Q4** The influence of different pretraining of the backbone is not properly assessed.
> >
> >  > **A4** Our framework is architecturally versatile, seamlessly integrating with a range of established transformer backbones such as PVT-v1, PVT-v2, DaViT, and FocalNet, as we have highlighted. This adaptability allows for extensive applicability across various computational paradigms. As shown in Figure 5, there is a clear trade-off between the complexity and size of the backbone and its performance. The larger the backbone, the better the performance. This relationship is crucial for users who must balance the computational demands with the performance requirements of their specific applications. Our findings indicate that opting for a more extensive backbone yields superior performance, providing a strategic advantage for those seeking to maximize the efficacy of their models.
> >
> >  **Q5** The authors claim that most semantic segmentation models struggle to distinguish between glass and non-glass regions, but does this assertion still hold true for the state-of-the-art generic semantic segmentation model, such as SAM?
> >
> >  > **A5** It is important to note that the SAM model **does not include semantics**, though. As suggested by the reviewer, we have tried the SAM model for some scenes and the results are shown in Section D.4, Figure 15 of the revised version because we cannot include images on OpenReview.

---

### Author Response · Authors · 2023-11-20
**Overall response**

Dear reviewers,

We thank all the reviewers for their constructive feedback.

**All reviewers** agree on the extensive experiment of our method with good performance on multiple datasets, and on the well-organized paper writing.
The reviewers also appreciate the well-established motivation of our method (Reviewer WwQc) and its network design (Reviewer 3ba2).
We have provided responses to the major concerns raised by each reviewer.

Following suggestions from the reviewers, we have the following changes (with text highlighted in blue in the revised version of our paper):

1. Section 2 now includes the latest research with a restructured layout for improved clarity.
2. We condensed the explanations of Position Embedding, and Encoder and Decoder in Section 3.1 for brevity.
3. Section 3.3 has been expanded with further details, accompanied by an updated Figure 7.
4. We show more failure cases in Figure 4, enhancing our discussion on system limitations.
5. Table 6 now features FLOPs and Params metrics for each configuration, providing a deeper insight into the computational efficiency of our models.
6. Section D.4 has been enhanced with new experiments, including supporting texts, Table 20, and Figure 15, to give a more comprehensive view of the Generic Segmentation capabilities.

Could the reviewers please let us know if there are other concerns after reading our responses? We will be happy to address any further concerns.

Regards,

The authors

---

### Author Response · Authors · 2023-11-22
**Follow-up on rebuttal and a kind reminder**

Dear reviewers,

We express our gratitude to all the reviewers for their constructive comments and careful reviews, as they significantly contributed to enhancing our article.

In continuation of our response, we would like to cordially remind the reviewers that the deadline for concluding the discussion is approaching. This open response window aims to facilitate a discussion regarding the paper, address any further inquiries, and enhance the overall quality of our work. Therefore, have you been able to peruse the responses provided below, wherein we have diligently tried to address your concerns? We aim to ascertain that the reviewers have deemed our responses solid and persuasive. We are willing to provide more information or explanation upon request.

Regards,

The authors

---

### Meta-Review · Area_Chair_ZhbB · 2023-12-10

**Metareview:**

This paper proposes to enhance semantic transparent object segmentation using a pyramid vision transformer with a Boundary Feature Enhancement (BFE) module (which features a boundary loss) and a Reflection Feature Enhancement (RFE) module (which features a reflection loss and reflection area decomposition).  The proposed work is benchmarked extensively not only on glass/mirror segmentation but also on generic semantic segmentation which includes both transparent and non-transparent object labels.

The strengths of the paper are the more effective use of low-level cues in transparent object segmentation and the extensive experimental validation.  The weaknesses of the paper are the ad-hoc nature of the techniques, a confusing array of baselines, and the lack of novel theoretical insights into transparent object segmentation.

The paper has received 3 detailed reviews and the authors respond with further clarifications and additional experimental results, which reviewers appreciate.  The final ratings are 6/5/5.  Based on the luke-warm reception of this paper, the AC recommends rejection.

**Justification For Why Not Higher Score:**

Incremental technical novelty and reviewers' luke-warm reception.

**Justification For Why Not Lower Score:**

N/A

---

### Decision · Program_Chairs · 2024-01-16

Reject